# A multi-step nucleation process determines the kinetics of prion-like domain phase separation

Erik W. Martin [1✉], Tyler S. Harmon[2], Jesse B. Hopkins [3], Srinivas Chakravarthy[3], J. Jeremías Incicco[4,5], Peter Schuck [6], Andrea Soranno [4,5] & Tanja Mittag [1✉]

Compartmentalization by liquid-liquid phase separation (LLPS) has emerged as a ubiquitous mechanism underlying the organization of biomolecules in space and time. Here, we combine rapid-mixing time-resolved small-angle X-ray scattering (SAXS) approaches to characterize the assembly kinetics of a prototypical prion-like domain with equilibrium techniques that characterize its phase boundaries and the size distribution of clusters prior to phase separation. We find two kinetic regimes on the micro- to millisecond timescale that are distinguished by the size distribution of clusters. At the nanoscale, small complexes are formed with low affinity. After initial unfavorable complex assembly, additional monomers are added with higher affinity. At the mesoscale, assembly resembles classical homogeneous nucleation. Careful multi-pronged characterization is required for the understanding of condensate assembly mechanisms and will promote understanding of how the kinetics of biological phase separation is encoded in biomolecules.

[1] Department of Structural Biology, St. Jude Children's Research Hospital, Memphis, TN, USA. [2] The Max Planck Institute for the Physics of Complex Systems, Dresden, Germany. [3] The Biophysics Collaborative Access Team (BioCAT), Department of Biological Sciences, Illinois Institute of Technology, Chicago, IL, USA. [4] Department of Biochemistry and Molecular Biophysics, Washington University in St. Louis, St. Louis, MO, USA. [5] Center for Science and Engineering of Living Cells (CSELS), Washington University in St. Louis, St. Louis, MO, USA. [6] Dynamics of Macromolecular Assembly Section, Laboratory of Cellular Imaging and Macromolecular Biophysics, National Institute of Biomedical Imaging and Bioengineering, National Institutes of Health, Bethesda, MD, USA. ✉email: erik.martin@stjude.org; tanja.mittag@stjude.org

Cellular function requires the organization of biomolecules in time and space. In bacteria[1,2] and eukaryotes[3], the selective condensation of subsets of biomolecules through liquid–liquid phase separation (LLPS) provides a means of spatial and temporal organization[4–9]. Importantly, these condensates can rapidly form and dissolve in response to gradients in biomolecule concentration[7] or environmental stimuli[10,11]. Effort has been devoted to understanding the environmental conditions at which biomolecular condensates form at equilibrium[12–14]. However, compartmentalization by LLPS is often used in higher organisms when a dynamic response or precise timing is required, such as in the stress response[10,15], when amplifying signals[16,17], or assisting in endocytosis[18,19]. For example, the nucleolus takes advantage of the plasticity inherent in phase-separated compartments to adapt in shape and size to its environment and rapidly dissolve and reform during cell division[7]. Therefore, the time evolution of the assembly of biomolecular condensates is another important parameter that determines biological function, and mechanisms may have evolved to control it. To start to understand the determinants of phase separation kinetics, the work presented herein will focus on the non-equilibrium process by which phase separation is nucleated from a homogenous solution.

Liquid–liquid phase separation occurs in a solution of biomolecules when the balance of interactions between biomolecules, and between biomolecules and the solvent, favor the decomposition of the solution into two or more distinct phases with different concentrations. The saturation concentration ($c_{sat}$) of a biomolecule is defined as the equilibrium concentration above which the solution transitions from homogeneous to phase-separated. A coexistence curve, or binodal, maps the saturation concentration and the coexisting dense phase concentration as a function of solution condition (e.g., temperature, pH, ionic

strength) (Fig. 1). The binodal directly reports on the free energy surface of the solution at equilibrium.

A solution of biomolecules can be rapidly perturbed, or "quenched", from the one-phase into the two-phase regime such that the system does not have time to equilibrate. The concentration of the quenched solution is therefore instantaneously above $c_{sat}$, homogeneous, and out of equilibrium. How far the solution is from equilibrium is controlled by how far the solution conditions are perturbed into the two-phase regime, or the "quench depth". At a given quench depth, the difference between the actual solution concentration and $c_{sat}$ is referred to as the degree of supersaturation ($\sigma$).

If the solution is quenched into the spinodal regime (Fig. 1), energetically spontaneous phase separation occurs. Spinodal decomposition represents the "speed limit" of the rate at which molecules can diffuse into dense phase droplets. The area in the two-phase region between the binodal and the spinodal boundaries is metastable, i.e., phase separation is not spontaneous. Phase separation in the metastable region requires nucleation (Fig. 1) akin to supercooled water that does not freeze until nucleated. Most biologically relevant phase separation is expected to follow a nucleation mechanism[20,21].

According to classical homogeneous nucleation theory, clusters of biomolecules form due to thermal fluctuations. The free energy associated with the formation of a cluster of a defined radius, $R$, is given by

$$\Delta G_{\mathrm{cluster}}(R) = 4\pi R^2 \gamma + \frac{4}{3}\pi R^3 \epsilon \qquad (1)$$

The free energy of forming a cluster depends on the surface tension, $\gamma$, and the free energy per unit volume, $\epsilon$, of adding a molecule to a cluster[22]. The surface energy is always unfavorable and will scale with the surface area of the cluster. The volume energy can be positive or negative. It is favorable as long as phase separation is thermodynamically preferred (i.e., above the saturation concentration) and will scale with the volume of the cluster. Above the saturation concentration, there is a competition between the favorable energy per volume and unfavorable energy per surface area. A critical size exists, termed the nucleation barrier, where both terms balance and the free energy is at a maximum. Clusters below this size tend to shrink, and clusters above this size tend to grow (Fig. 1). The difficulty in overcoming this barrier is strongly dependent on the favorability of the volume energy, which, in turn, is strongly dependent on the quench depth. The rate at which nucleation occurs is determined by the probability of reaching the critical cluster size and thus should strongly depend on the quench depth.

Whether LLPS of proteins proceeds through homogenous nucleation as described above or whether additional distinct assembly steps precede nucleation is currently unclear. LLPS of some proteins occurs quickly, whereas other proteins, such as α-synuclein and tau, seem to require hours to form a dense phase[23–25]. These different nucleation rates may purely result from different quench depths, but given that the kinetics of protein phase separation has not been carefully determined as a function of quench depth, this currently remains unclear. Alternatively, the different nucleation rates may point to a role of slow conformational changes or oligomerization for the formation of critical nuclei. Pathological aggregation of several proteins is known to be preceded by the formation of distinct oligomers[26–30]. Similarly, LLPS assembly routes that include oligomerization may be common, and their timing may be easier to control via post-translational modification or via the availability of binding partners than pure homogenous nucleation. How nucleation of protein phase separation occurs on molecular length scales from single molecules to small oligomers, i.e., the

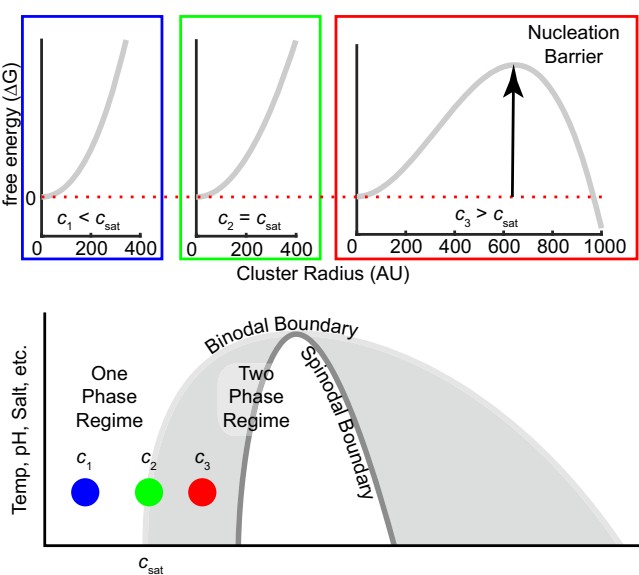

**Fig. 1 The free energy barrier to nucleation depends on the degree of supersaturation.** The free energy as a function of cluster size is shown on the top for three concentrations, i.e., below, at and above the saturation concentration. As the concentration increases from subsaturated to the saturation concentration to supersaturated, the sign of the free energy difference between a molecule inside and outside a cluster flips from positive to negative, and a finite nucleation barrier emerges (Eq. (1)). The gray region inside the binodal is metastable, meaning that nucleation is required to form dense phase clusters that grow until equilibrium is reached. The white regime inside the spinodal is unstable, meaning the solution spontaneously decomposes into dilute and dense phases.

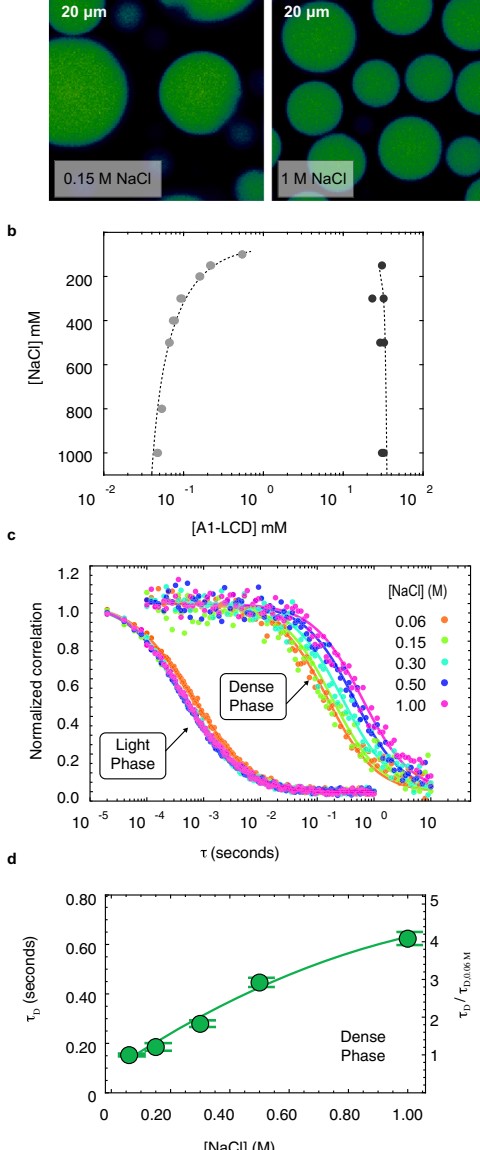

**Fig. 2 NaCl promotes equilibrium phase separation of A1-LCD. a** Confocal microscopy images of fluorescently labeled A1-LCD droplets at 0.15 M NaCl and 1 M NaCl. Many fields of view from two independent replicates were examined with consistent results. Images are displayed with false-color determined by fluorescence intensity. **b** A1-LCD binodal mapped by UV absorption concentration measurements of coexisting dilute and dense phases as a function of NaCl concentration. The saturation concentrations at each NaCl concentration were measured independently for three different samples and all individual data points are displayed (light gray, overlapping). Dense phase concentrations for each NaCl concentration were measured from two different samples (dark gray). The gray dashed line serves to guide the eye. **c** Normalized correlations in the dilute (left) and dense phases (right) as a function of NaCl concentration. Experimental conditions were the same as above but in presence of 1 mM BME. **d** The FCS lag time inside droplets ($\tau_D$ in Supplementary Eq. 10) as a function of NaCl concentration. The right axis shows the relative value with respect to the value measured at 60 mM NaCl, reflecting the change in apparent viscosity due to the salt concentration. The green line is a fit to a second-order polynomial. Two measurements from two independent droplets were averaged and used to fit $\tau_D$. Error bars represent the standard error of the fit.

nanoscale, and on the order of clusters of many molecules with radii over 100 nm, i.e., the mesoscale, is thus an interesting question.

In this work, we characterize nucleation of in vitro phase separation of a prototypical prion-like domain, the low-complexity domain of hnRNPA1 (A1-LCD) as a model phase-separating protein. Previous work has quantitatively characterized equilibrium A1-LCD phase separation[4,31]. We now apply novel rapid-mixing, time-resolved X-ray scattering (TR-SAXS) methods to reveal the kinetics of protein cluster formation in real-time on the nanoscale in supersaturated solutions. We combine these measurements with the characterization of the solution composition at equilibrium to determine how clusters of A1-LCD form and how these clusters lead to phase separation.

We show that phase separation of A1-LCD seems to proceed via a homogeneous nucleation mechanism at the mesoscale, i.e., if we consider particles above a certain size. The barrier to nucleation and hence the rate of phase separation is determined by protein concentration and ionic strength (and presumably additional parameters that control the quench depth including temperature, counterion identity, and pH). At early time steps at the molecular or nanoscale, the size distribution of small A1-LCD oligomers deviates significantly from what is expected from classical nucleation theory. This deviation has a large impact on the nucleation rates, and the molecular details of small oligomer formation must be taken into account to describe the kinetics of phase separation accurately. Deconvoluting the effects at these length scales requires monitoring the phase separation process from the nanoscale to the mesoscale. Our insights may explain why phase separation of different biomolecules happens over vastly different timescales.

## Results

**Salt concentration controls the driving force of hnRNPA1 LCD phase separation.** We have previously characterized the equilibrium phase behavior of A1-LCD as a function of temperature and have shown that it is well approximated by classic homo-polymer mean-field models[31]. Similar to other prion-like domains[32], A1-LCD phase separation is enhanced with the addition of monovalent salts[33] and NaCl can be used as a parameter to control quench depth. The addition of NaCl leads to robust phase separation (Fig. 2a). The saturation concentration ($c_{sat}$) and corresponding dense phase concentration were measured as a function of NaCl concentration (Fig. 2b) to construct the complete binodal. $c_{sat}$ decreases strongly with increasing NaCl concentration, particularly between 50 and 500 mM NaCl. The precise mechanism of how salt impacts phase separation is a subject of active study and is likely protein-dependent. In the case of A1-LCD, we suspect that increasing ionic strength screens repulsive interactions originating from the net positive charge. As the salt concentration increases further, the enhancement of hydrophobic interactions promotes distributive interactions between aromatic residues[34,35]. These effects depend on the specific salt type[34,36]. To probe the properties of the dense phase, we measured the diffusion time of A1-LCD molecules inside droplets by fluorescence correlation spectroscopy (FCS) (Fig. 2c, d). Rescaling the measured diffusion times by the value at the lowest salt concentration provides an estimate of the relative mobility inside the droplet. The apparent viscosity increases, but the protein concentration in the dense phase is largely constant, pointing to stronger intermolecular interactions in the resulting dense phase, i.e., stronger networking. Together these data indicate that the driving force for phase separation increases with the NaCl concentration.

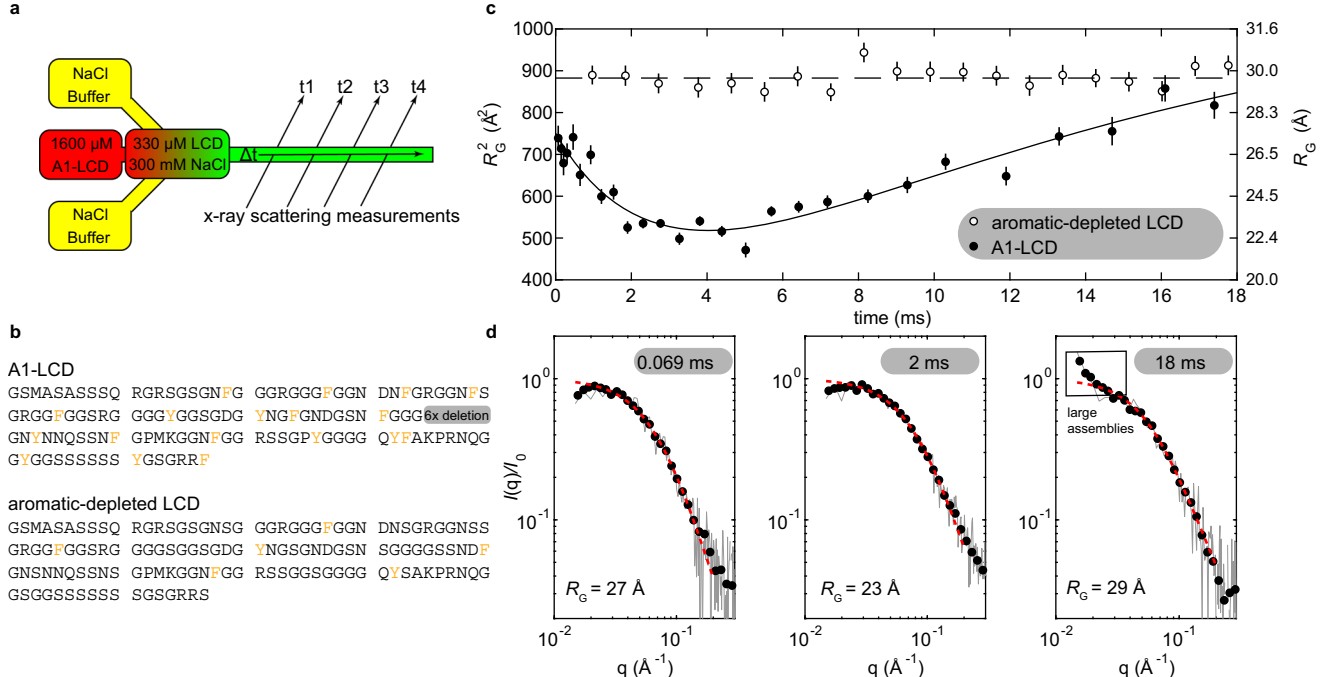

**Fig. 3 Chaotic-flow time-resolved SAXS experiments reveal the collapse of A1-LCD before its assembly. a** A schematic showing the mixing of A1-LCD (initially without excess salt in the one-phase regime, red) with NaCl-containing buffer (yellow). The resulting solution is in the two-phase regime (green). **b** The amino acid sequences of A1-LCD and the aromatic-depleted LCD. Aromatic amino acid residues are colored orange. **c** $R_G$ and $R_G^2$ of the A1-LCD (black) and an aromatic-depleted variant of the A1-LCD (open circles). The $R_G$ values are calculated by fitting the IDR form factor to data binned for 6 consecutive frames. The solid line is a fit to the sum of exponential collapse ($\tau \sim 470\ \mu s$) and growth ($\tau \sim 36\ ms$) (Supplementary Eq. 1). The dashed line is a horizontal line at the mean value for the aromatic-depleted variant. Error bars represent the standard error in the fit to the IDR form factor. **d** Individual SAXS curves from points along the time course shown in **b**. Raw curves were selected to represent the initial state after full mixing, the minimum $R_G$, and the end of the observable time window. Black dots are data logarithmically smoothed into 30 bins, gray lines are raw data, and red dashed lines are fits to the IDR form factor from which $R_G$ was extracted. The boxed region at 18 ms highlights the departure of the form factor at small angles from a monomer and thus indicates beginning assembly, beyond which point we stop reporting $R_G$ values.

For flexible homopolymers, the driving force for phase separation and the coil-to-globule transition are linked via a well-explored theoretical relationship[37]. The existence of this connection in protein systems has been demonstrated both computationally and experimentally[20,31,38,39]. We have previously shown that the single-chain dimensions of A1-LCD (as measured by the radius of gyration, $R_G$) are predictive of phase-separation boundaries, and both can be manipulated by changing the number of aromatic amino acids in the sequence[31]. We have also shown that the $R_G$ decreases with increasing NaCl concentration[33], and that $c_{sat}$ decreases as well (Fig. 2b, Supplementary Fig. 1). These results suggest an equivalence between inter- and intramolecular interactions reminiscent of homopolymers and validate the use of NaCl concentration as a control parameter. Therefore, we expect the response to quenching into the two-phase regime to be a decrease in monomer $R_G$ followed by assembly into clusters and finally phase separation.

**Sub-millisecond timescale collapse of A1-LCD.** To determine changes in single-chain dimensions of A1-LCD upon quenching the solution into the two-phase regime, we used time-resolved small-angle X-ray scattering (TR-SAXS) experiments that take advantage of chaotic-flow mixing. The NaCl concentration was quenched from no excess salt to 300 mM NaCl (Fig. 3a) and the mixer allows for measurements as early as 69 μs after quenching. These measurements showed compaction of A1-LCD with a decay constant of ~470 μs (Fig. 3b). We used the same chaotic-flow mixer to record the response of cytochrome c to dilution of denaturant, a well-characterized refolding process, to ensure the

mixer was properly functioning. The measured time constants were consistent with the established kinetics of refolding (Supplementary Fig. 2)[40]. Interestingly, the time constant of A1-LCD collapse is of the same order as the initial step in cytochrome c refolding, which has been attributed to a barrier limited collapse, suggesting that the reorganization of A1-LCD to more compact conformations might also be barrier limited.

We wondered whether A1-LCD collapse at early mixing times resulted from partial folding and thus compared the shape of individual SAXS profiles at the earliest time points and at the minimum dimension. The shapes were similar and overlaid well when normalized by $I_0$ and $R_G$, indicating that reduction in size results from a slight decrease in the average dimensions, not from complete or partial folding (Fig. 3c and Supplementary Fig. 3). A variant of the A1-LCD that has only one-third of the aromatic residues and thus does not undergo phase separation even at high concentration and low temperature[31] showed no sign of collapse or assembly on the measured timescale (Fig. 3b).

It is worth noting that we observed an increase in $I_0$ associated with the $R_G$ decrease (Supplementary Fig. 4). The effect is repeatable among independent experiments with A1-LCD but does not appear with the aromatic-depleted LCD or cytochrome c (Supplementary Fig. 2b), ruling out mixing artifacts. An increase in $I_0$ is often associated with an increase in mass, but the clear decrease in $R_G$ as measured at very small angles coupled with the absence of any major change in the shape of the form factor, rule out folding or oligomerization of a fraction of the sample as an explanation for this observation. We instead suggest that compact conformations of disordered proteins may have more stable

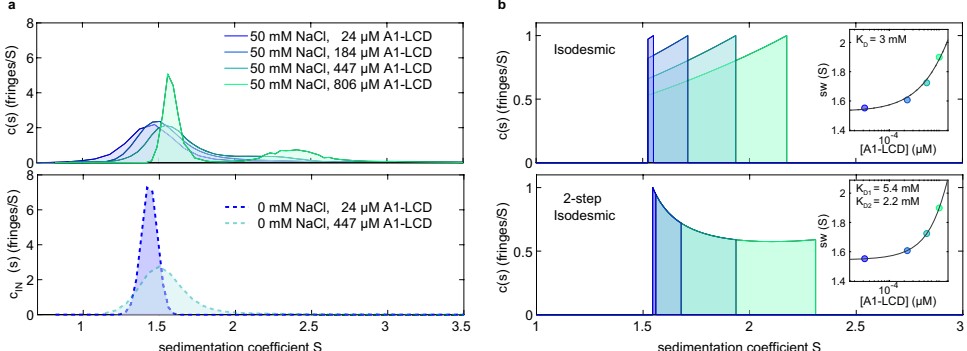

**Fig. 4 The A1-LCD forms small oligomers in the dilute phase. a** Normalized SV-AUC sedimentation coefficient distributions as a function of A1-LCD concentration. In samples with 50 mM NaCl, and approaching the saturation concentration, a second peak appears, shifts to higher molecular weight, and increases in intensity, implying that the size distribution of A1-LCD molecules shifts from mostly monomer to encompass also higher-order assemblies. **b** An isodesmic assembly model poorly accounts for the shape of the sedimentation profile as a function of concentration. The weight-averaged sedimentation coefficient as a function of concentration is fit to an isodesmic model (inset). A model with initial low-affinity association followed by higher-affinity addition of monomers qualitatively agrees with the experimental data. The weight-averaged sedimentation coefficient as a function of concentration is fit to a two-step isodesmic model (inset). Colors in **b** correspond to colors in **a**.

hydration layers. While a collapsed IDR has a less solvent accessible area, we posit an increase in stability of the hydration layer which would, in turn, increase contrast and $I_0$ while decreasing $R_G$ (Supplementary Fig. 5 and Supplementary Note 1).

**A1-LCD forms clusters at conditions near the binodal boundary.** The final condition of the chaotic-flow mixing experiment (300 mM NaCl and 0.33 mM A1-LCD), after equilibration, places the A1-LCD solution into the two-phase regime at equilibrium (Fig. 2b). At ~2 ms after mixing, the $R_G$ reaches a minimum and begins to increase (Fig. 3c). SAXS profiles show the evolution of interparticle interference or the presence of larger assemblies as indicated by an upturn at small angles. The appearance of this upturn (Fig. 3d, box) demonstrates that the apparent increase in $R_G$ is due to assembly and not simply an expansion of a single chain. Given the slow evolution of the form factor and the fact that large assemblies do not appear until ~18 ms after mixing, it is unlikely that phase separation is diffusion-limited. It must therefore occur via a nucleation pathway.

To obtain an orthogonal view of the size distribution of A1-LCD clusters at protein concentrations approaching $c_{sat}$, we used equilibrium velocity analytical ultracentrifugation (SV-AUC). At the highest protein concentrations characterized in the presence of NaCl, the solution is slightly above the saturation concentration. Any dense phase droplets that formed sedimented rapidly under these conditions and the resulting measurements were poised at $c_{sat}$. In the absence of excess NaCl, A1-LCD exhibited strongly nonideal sedimentation (best estimate $k_S = 28$ mL/g) with little indication of oligomerization (Fig. 4a). In contrast, sedimentation of samples containing 50 mM NaCl showed markedly decreased nonideality and a clear concentration-dependent shift in the sedimentation velocity distribution peaks, characteristic of self-association (Fig. 4a). To characterize the formation of small assemblies that form on-pathway to the formation of critical nuclei, we assumed that simple association models would be valid in the limit of clusters containing only a few molecules. The weight-average sedimentation coefficient as a function of concentration can be modeled well theoretically, but multiple assembly models resulted in good agreement with the data (Fig. 4b, insets). Using Gilbert theory[41,42], which describes asymptotic boundaries of self-associating systems in rapid exchange and in the absence of diffusion, we simulated the theoretical $c(s)$ distributions to assess the validity of different assembly mechanisms. The bimodal nature of the $c(s)$

distributions at higher concentrations is qualitatively inconsistent with either monomer/dimer self-association or isodesmic self-association, which both result in monomodal shapes with a maximum at the highest $s$-value (Fig. 4b). However, the bimodal nature of the sedimentation coefficient distribution matches an oligomerization scheme in which an initial weak dimerization step is followed by a subsequently more favorable addition of molecules (Fig. 4b).

**Nucleation is slow near the binodal boundary.** To examine how mesoscale assemblies develop over longer timescales, we used laminar-flow microfluidic TR-SAXS experiments that extended the measurement window to 80 milliseconds but lack the ability to observe microsecond timescales. For laminar-flow experiments, the samples were projected into a thin stream into which $Na^+$ and $Cl^-$ ions diffuse from the surrounding buffer, resulting in only minimal dilution of A1-LCD samples after injection (Fig. 5a). This approach enabled access to higher A1-LCD concentrations of ~0.6 mM (0.598 mM, i.e., 7.5 mg/mL) across NaCl concentrations ranging from 200 to 500 mM and thus a greater degree of supersaturation (Fig. 5b).

The flow rate in the microfluidic mixer is fast relative to the SAXS exposure time, and each molecule is only observed on the order of milliseconds. Each data point represents a snapshot of many molecules (~3.4 × 10^13, Supplementary Methods) at an identical time after mixing. The flow rate of the sample is higher than the scan rate of the X-ray beam along the stream, and the ensemble of molecules at each data point is therefore uncorrelated with the ensemble at all other data points. Variation in the SAXS form factor between data points at a given time after mixing thus represents stochastic fluctuations in the size of clusters in the sample.

The SAXS experiments are sensitive to fluctuations on two distinct length scales. (1) In supersaturated samples in the absence of critical nuclei, dimers and small oligomers form, and we assume that they are similar to those observed in the subsaturated AUC experiments (Fig. 5c). On this length scale, the SAXS curve is the result of the mass average of the distribution of oligomeric species, and the ensemble $R_G$ is within the measurable range of the experiment (i.e., $R_G$ is smaller than the maximum resolvable $R_G$, which is approximately the inverse of the smallest measured angle, i.e., 10–13 nm). (2) Mesoscopic assemblies rapidly form in supersaturated samples in which clusters of A1-LCD exceed the critical nucleation size. These assemblies exceed

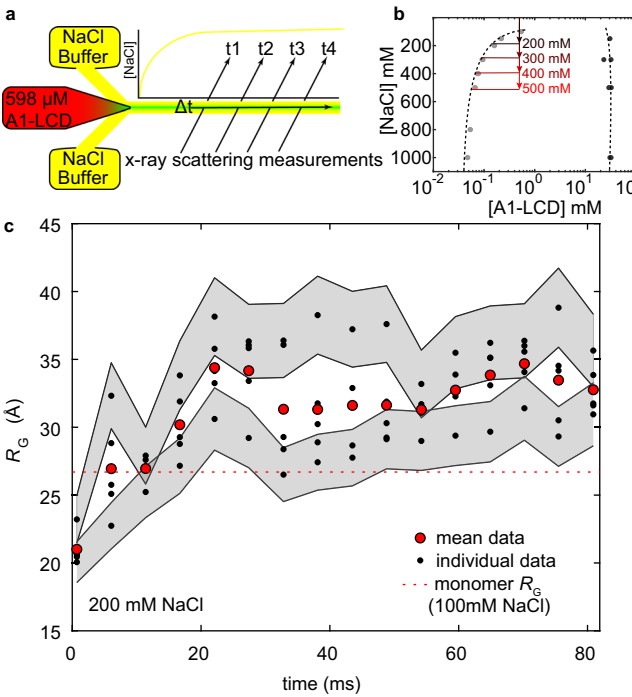

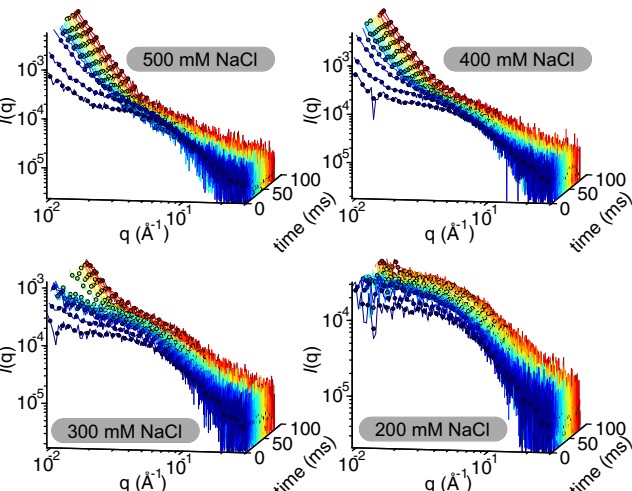

**Fig. 6 The time evolution of the form factor in laminar-flow TR-SAXS experiments.** Raw data is displayed for all samples as lines and logarithmically smoothed data as filled circles. Curves from early time points are shown blue, from late time points red, with a color gradient to indicate time evolution for in-between time points. The first recorded curve is shown to represent each time point.

**Fig. 5 Laminar-flow time-resolved SAXS experiments. a** A schematic showing the laminar-flow experiment. A protein sample without excess salt (red) is focused into a thin stream, and ions from a sheath of buffer (yellow) rapidly diffuse into the solution, resulting in a sample of protein in a NaCl-containing solution (green). Scattering data is recorded as a function of position from mixing; data collection begins after the initial fast mixing phase. Data from successive time points are uncorrelated because the scan rate of teh X-ray beam is slower than the flow rate of the protein stream. The dead time is defined by the period required for a threshold of 50% mixing to occur. **b** The quench depth of individual laminar-flow TR-SAXS experiments depicted on the A1-LCD binodal (from Fig. 2b). **c** $R_G$ values as a function of time after mixing for a sample mixed into 200 mM NaCl. All individual measurements are shown as black dots. Red circles are the average of all measurements at a particular time after mixing, i.e., position on the mixer. The uncertainty bounds represent the standard deviation in determining the $R_G$ from Guinier analysis[78]. The standard deviation is similar for each data point, and thus, gray shaded regions are centered only on the maximum and minimum $R_G$ values at each time point for clarity.

the maximal experimentally measurable dimension and the assemblies manifest as a power-law decay at small angles in the SAXS curve (Supplementary Note 2). A large ensemble of molecules is measured at each time point, and individual molecules, or clusters of molecules, are only observed for a fraction of the integration time. Therefore, we cannot observe subcritical clusters growing and reaching critical size in the SAXS experiment. Instead, the data either reports on the distribution of subcritical clusters in the absence of nucleation or effectively counts the fraction of the system in mesoscopic assemblies in the presence of nucleation.

Data measured at the shallowest quench depth (200 mM NaCl) does not show evidence of significant mesoscopic assembly on the timescale of the experiment. At the earliest time points, the $R_G$ is similar to the minimum radius measured in the chaotic-flow experiments and smaller than equilibrium measurements (Fig. 5b). The radius increases over the first 20 ms in agreement with the timescale of the chaotic-flow experiment. After 20 ms, the mean $R_G$ does not appreciably change but fluctuates around a value consistently larger than the monomer. The magnitude of

the cluster size fluctuations between data points is approximately one order of magnitude greater than the uncertainty (Fig. 5c). Higher molecular weight clusters have a greater contribution to the scattering profile, and fluctuations in measured $R_G$ likely arise from small changes in the population of relatively rare, larger clusters. While this solution is supersaturated, the data are consistent with a subcritical distribution of clusters. We assume that the size distribution of oligomers is consistent with that observed in subsaturated solutions characterized by AUC.

**The nucleation rate is determined by the quench depth.** Mesoscopic assemblies rapidly appeared in the sample volume when the NaCl concentration was greater than 200 mM (Fig. 6). The power-law decay at small angles resulting from mesoscopic assemblies prevents accurate determination of $R_G$. We, therefore, adopted a procedure to follow the volume fraction of mesoscopic assemblies. The earliest time points in the experiment were fit to a Gaussian chain form factor. The evolution of the structure factor was followed by calculating an "assembly metric" which is the difference in intensity at the smallest measured angles between the experimental and the calculated monomer form factor. The assembly metric reports on the total volume of all clusters (Supplementary Fig. 6 and Supplemental Note 2).

The rate of mesoscopic assembly is strongly dependent on the quench depth at NaCl concentrations ranging from 300 to 500 mM (Fig. 6). At late times and high NaCl concentrations, the form factor approaches the form factor of a homogeneous dense phase measured at equilibrium conditions (Fig. 6 and Supplementary Fig. 7). We thus conclude that laminar-flow TR-SAXS experiments capture the formation of mesoscopic dense protein droplets.

We used a Weibull probability distribution function (PDF) to determine the rate of nucleation as a function of quench depth. The Weibull PDF is a phenomenological equation that can describe the probability of a stochastic event happening as a function of time and has been used to analyze nucleation of crystallization and amyloid formation[43,44] The shape of the Weibull distribution matches the time evolution of the mean degree of assembly at 300–500 mM NaCl (Fig. 7a–c). The peak in the probability distribution indicates the time point with the

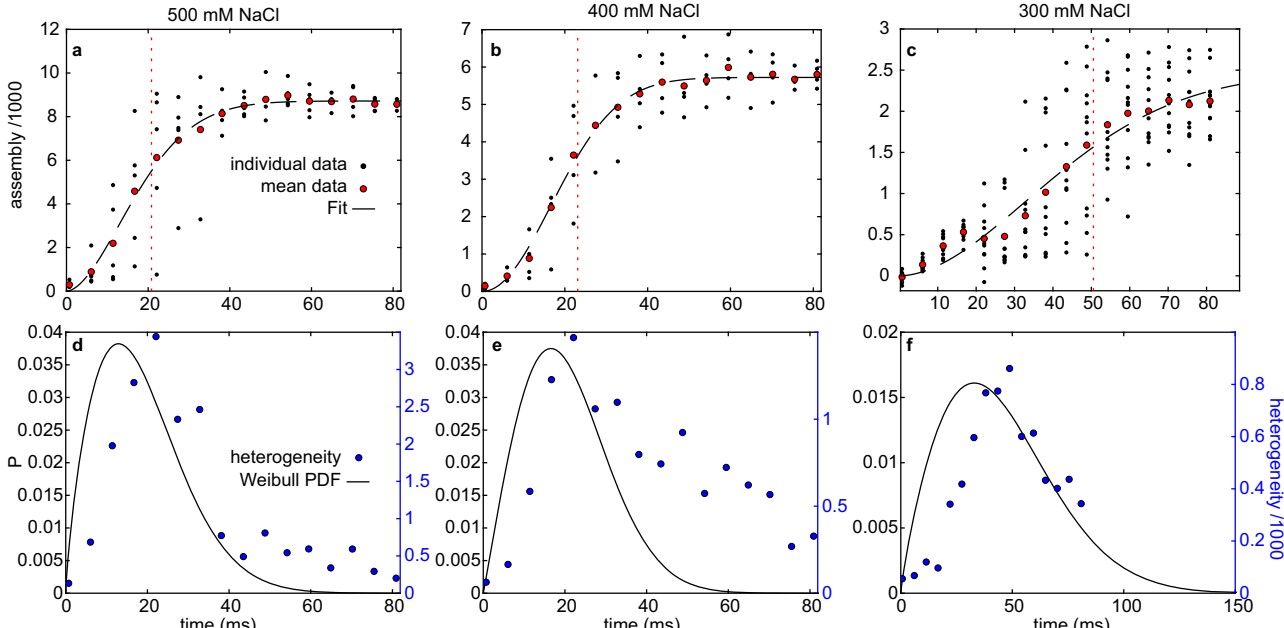

**Fig. 7 The quench depth determines nucleation and assembly rate. a–c** Assembly metric calculated from the SAXS structure factor as a function of time for different quench depths (i.e., NaCl concentrations). Black dots are individual measurements, and red circles are the average of all measurements at each time point. The black dashed line is a fit to a scaled Weibull cumulative probability distribution. The horizontal red dotted line shows the mean of the related PDF. **d–f** Fit to the Weibull distribution transformed into a probability density distribution. The blue circles are the standard deviation of the distribution of individual data points in **a–c**.

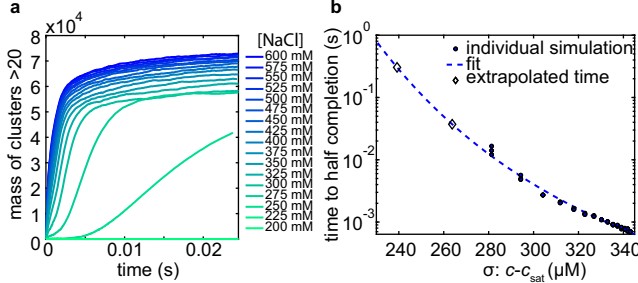

**Fig. 8 Simulations of A1-LCD classical homogeneous nucleation rates.**
**a** Nucleation is represented as the mass of clusters in the simulation with greater than 20 monomers versus simulation time. Each trace is the average of three independent simulations. **b** The time at half completion in **a** is used as a proxy for nucleation time and is plotted versus the degree of supersaturation in the simulation.

largest variance in the probability of a nucleation event and should therefore exhibit the highest heterogeneity in the assembly metric. Indeed, the probability density function calculated from the fit parameters superimposes well with the standard deviation of individual data points, indicating that the variance in the assembly metric captures the stochasticity of nucleation (Fig. 7d–f). This variance is also clear from comparisons of raw data recorded at the same time interval after mixing; the form factor of individual measurements near the peak of the probability density function also varies widely (Supplementary Fig. 8).

**Mesoscopic A1-LCD assembly seems to resemble homogeneous nucleation.** Motivated by the hypothesis that phase separation of A1-LCD follows a homogeneous nucleation mechanism (Fig. 1), we developed an analytic model describing the kinetics of this

process. Our model incorporates the experimentally determined dilute and corresponding dense phase concentrations ($c_{sat}$ and $c_{den}$, respectively, Fig. 2b) as well as the diffusion coefficient of monomeric A1-LCD as measured by FCS (Fig. 2c, d and Supplementary Fig. 9). The following expression describes the flux of molecules between the dilute and dense phases:

$$\dot{N} = -DRc_{sat}\left(1 - \frac{2\gamma}{Rc_{den}}\right) + DRc \qquad (2)$$

where $R$ is the droplet radius, $\gamma$ is the surface tension, and $c_\infty$ is the concentration of protein in the bulk phase at a given time. This expression can be used to compose a stochastic integrator which describes the probability of a cluster growing or shrinking (see Supplementary Methods). Using these numerical simulations, we explored the expected rate with which droplets appear in solution as a function of $c_{sat}$ in the limiting case of homogeneous nucleation. The assembly rate increases with the degree of supersaturation (σ) as expected. In this model, σ is increased by decreasing $c_{sat}$, which corresponds to increasing concentrations of NaCl in the experiment (Fig. 8 and Supplementary Figs. 10, 11).

We conclude that A1-LCD phase separation kinetics can be described by a homogeneous nucleation pathway at the mesoscopic scale based on a comparison of experimental data with numerical simulations. The time to half assembly is exponentially dependent on the free energy barrier to nucleation (Fig. 8), which depends on the quench depth σ according to $\Delta G \sim e^{\sigma^{-2}}$ (Supplementary Note 3). The rates derived from the Weibull cumulative probability distribution are the experimental analog to the time to half assembly from the simulations. A similar dependence is observed for the data points between 300 and 500 mM NaCl (Fig. 9a) in experiments and simulations. In simulations performed at 200 mM NaCl, near the binodal boundary, the formation of mesoscopic clusters is not observed on the timescale of the simulation (Fig. 8). Similarly, parameters

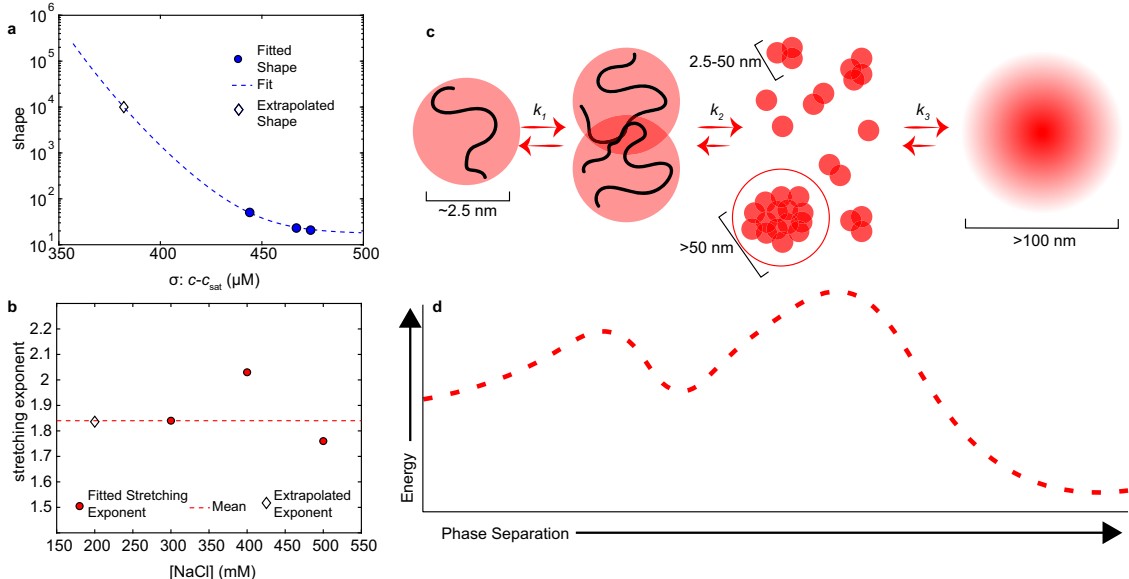

**Fig. 9 Mesoscopic A1-LCD assembly is nucleated by an activated state. a** The shape parameter defines the Weibull distribution width, which is the timescale over which the transition occurs, and it depends on the degree of supersaturation ($\sigma$). Using the values from the fits in Fig. 7, the shape parameter for 200 mM NaCl can be extrapolated (diamond). **b** The stretching exponent varies minimally as a function of NaCl concentration. The average of all values from 300 to 500 mM NaCl is indicated with a red dashed line. Given the absence of a clear NaCl dependence the average value is used to calculate a PDF for 200 mM NaCl. **c** Schematic of phase separation of A1-LCD proceeding via three steps. 1. Low-affinity dimers form. 2. Monomers are added onto low-affinity small clusters to form a distribution of species as predicted by classical nucleation theory. 3. A large cluster grows with low probability beyond a critical size (circled in red) and the resulting nucleus grows. At each step, the approximate length scale is given based on the measurable scales in SAXS experiments. **d** The schematic in **c** is drawn as a phenomenological reaction coordinate based on the known barriers to dimer formation and phase separation.

extrapolated from data at 300–500 mM NaCl predict a lack of assembly within the 80 ms laminar-flow TR-SAXS measurement window at 200 mM NaCl (Supplementary Fig. 12).

The small clusters observed by AUC at the saturation concentration are below the nucleation size and therefore disassemble again. In addition, the $K_D$ associated with initial complex formation is more unfavorable than that for further assembly (Fig. 4), pointing to an additional energy barrier to nucleation. The Weibull probability distribution contains a stretching exponent in addition to the rate, which reports on a lag time of assembly. In all laminar-flow TR-SAXS experiments, irrespective of $\sigma$, a stretching exponent of 1.8–2 was required to fit the data (Fig. 9a). Values of the stretching exponent greater than 1 indicate that the probability of nucleation increases as a function of time. The similar magnitude of the stretching exponent for all values of $\sigma$ suggests that a similar underlying process is required for nucleation that is independent of $\sigma$ (Fig. 9b). Taken together, we conclude that A1-LCD phase separation seems to follow a homogeneous nucleation pathway when observed on the mesoscale. However, careful characterization of initial assembly on the molecular or nanoscale shows that the pathway deviates from homogenous nucleation. The initial formation of small complexes is unfavorable, but once they are formed, they can grow effectively and nucleate phase separation (Fig. 9c).

### Discussion

In this work, we demonstrate that phase separation of A1-LCD on the mesoscale can be well described by a classic, homogeneous, nucleation-and-growth mechanism. However, we observe deviations from homogeneous nucleation on the nanoscale. Monomers assemble via unfavorable steps before the addition of additional monomers becomes favorable. Kinetic analysis of phase separation after quenching typically focuses on the rate of growth of the

droplet phase (ripening or coarsening)[21,45]. Thus, the fast time-scale behavior of the monomer in response to quenching into the two-phase regime is not observed. This behavior can be complex and requires the consideration of details on the molecular level.

The initial collapse of A1-LCD in chaotic-flow experiments occurs on timescales slower than expected based on chain reorganization times of typical IDPs[46,47]. The time constant ($\tau \sim 470$ μs) suggests that the collapse is barrier limited[40]. In a low NaCl buffer, the A1-LCD is more compact than a Gaussian chain ($\nu^{app} < 0.5$, Supplementary Fig. 1) which implies the presence of numerous favorable intramolecular contacts. This condition may be analogous to a metastable intermediate on the pathway to folding in a foldable protein. In foldable proteins, native-like contacts begin to form under conditions where $\nu^{app} \sim 0.5$[48,49]. Unlike folded proteins, disordered proteins lack a single stable conformation, and an ensemble of interconverting conformations containing a dynamic network of semi-stable contacts exists. Short- and long-range contacts can dramatically reduce the chain reorganization time[50,51]. In the case of A1-LCD, this manifests in the previously reported NMR transverse relaxation rates and NOEs that reflect semi-persistent contacts between aromatic amino acids[31]. Given our prior knowledge of this system, the initial collapse of the A1-LCD is likely characterized by an increase in the lifetime of multiple contacts (associated with energetic barriers) that give rise to a rough energy landscape and high internal friction, as opposed to the classical view of a single barrier between two states.

It is known that biomolecules can form higher-order assemblies and clusters below the saturation concentration. This has been experimentally observed in vitro for high concentration solutions of globular proteins[52,53], which can exhibit multi-step nucleation[54]. Clustering has additionally been observed for phase-separating domain-motif systems[55] and in elastin-like polypeptides[56,57]. Interestingly, complex micellular structures potentially nucleate phase separation in elastin-like

polypeptides[58]. Clustering has also been observed in aggregating systems in vivo[59]. The size distribution of subcritical clusters is determined by the nucleation barrier (Fig. 1) and is, therefore, easiest to characterize near the binodal boundary. Close to the boundary in the two-phase regime, we observe clusters forming in the first 20 ms of TR-SAXS experiments, but they do not grow into critical nuclei on the experimental timescale. Near the binodal boundary in the one-phase regime, the distribution of cluster sizes was quantified by SV-AUC. These data indicate that assembly of A1-LCD is a two-step process where initial clusters form with a high $K_D$ followed by the addition of monomers with a lower $K_D$. The initial unfavorable step generates a lag time in nucleation and effectively slows phase separation.

We monitored the evolution of phase separation from small clusters into mesoscopic assemblies in laminar-flow TR-SAXS experiments as a function of quench depth. These data are in qualitative agreement with analytic simulations of idealized homogeneous nucleation. Importantly, these data and simulations show that the nucleation time becomes asymptotically slow as the quench depth decreases (Fig. 9a). Solutions near the binodal may remain a single phase indefinitely in the absence of perturbations that aid nucleation, similar to supercooled water that does not freeze. The unfavorable initial association into small clusters differs from the analogy to freezing water. The combination of a high barrier to nucleation and potential unfavorable steps associated with small oligomer formation may explain why some phase-separating proteins, such as tau and α-synuclein, appear to require incubation periods on the order of hours or days prior to phase separation[23–25].

Insights gleaned from measuring the nucleation barrier of idealized systems in vitro are transferrable to biological systems. Notably, crossing a phase boundary alone is not enough to initiate rapid compartmentalization. The response time is linked to the probability of forming a cluster large enough to nucleate phase separation. Modulating the probability of nucleation provides biology with additional tunable parameters for regulation. In the simplest case, the concentrations of biomolecules could be rapidly increased, creating a highly supersaturated solution. For example, during the stress response, polysome disassembly rapidly frees up RNA[60] that can condense along with RNA-binding proteins into stress granules. Post-translational modifications (PTMs) may also provide mechanisms to modify the nucleation barrier. Modifications such as phosphorylation can increase[61,62] or decrease[63,64] the saturation concentration depending on the sequence context of the modified protein, effectively changing the degree of supersaturation. PTMs may alternatively change the nature of nanoscale assemblies. Alternately, biomolecules could be maintained in a supersaturated state if the nucleation barrier was high enough to prevent phase separation on relevant timescales, resulting in the system being kinetically trapped in the one-phase regime. By forming large high-affinity complexes upstream of phase separation that create a favorable surface for nucleation, the system could escape from the kinetic trap and proceed to a thermodynamically favorable state. This has been proposed for G3BP1 and UBAP2L complexes that form prior to stress granule formation[65,66] and this is likely a general mechanism. Many proteins are able to form higher-order assemblies. Glycolytic enzymes form filaments in response to changes in metabolite concentration[67]. These filaments, in turn, nucleate phase-separated puncta that recruit an array of metabolic proteins and RNPs[68]. An alternative heterogeneous nucleation pathway occurs along surfaces within the cell where favorable interactions with the surface decrease the energetic cost of nuclei formation, e.g., in membrane receptor signaling[16,17] or in endocytosis[18].

The results presented here stress that a complete understanding of phase separation of biomolecules requires both kinetic and equilibrium analysis. While the equilibrium quantities provide valuable information about the thermodynamics of phase separation, the kinetics of phase separation is equally critical for an understanding of how phase separation may mediate cellular function. By combining TR-SAXS experiments with equilibrium measurements, we have characterized how the quench depth impacts nucleation kinetics and have uncovered an early unfavorable step on the assembly pathway. Our results highlight that cells may regulate phase separation not just by modulating the saturation concentration but also by modulating the formation of complexes on the pathway to phase separation and hence modifying the rate of nucleation. Combining rapid-mixing TR-SAXS measurements with careful equilibrium characterization of oligomeric species has the potential to uncover the on-pathway steps to LLPS nucleation in complex mixtures where substantial deviations from homogeneous nucleation can be expected.

## Methods

**Protein purification.** A1-LCD and the aromatic-depleted LCD coding sequences (Supplementary Table 1) were synthesized and inserted into Gateway (Thermo-Fisher) expression vectors that code for an N-terminal poly-histidine affinity tag followed by a TEV protease cleavage site. Proteins were expressed in BL21 GOLD (DE3) cells using Terrific Broth rich media. The hnRNPA1 LCD variants expressed in inclusion bodies and were purified from the insoluble fraction. After cell lysis in a buffer containing 50 mM Tris pH 7, 300 mM NaCl and 2 mM β-mercaptoethanol (BME), inclusion bodies were collected by centrifugation and solubilized by addition of 6 M Gnd-HCl. Samples were clarified by centrifugation and further purified by Ni-NTA affinity chromatography using a buffer containing 50 mM Tris pH 7, 500 mM NaCl, and 4 M Urea. Ni-NTA wash steps included 15 mM imidazole and 50 mM imidazole. Protein was eluted using 300 mM imidazole. The polyhistidine tag was cleaved off by the addition of TEV protease while the sample was dialyzed against a buffer containing 50 mM Tris pH 7, 20 mM NaCl, 5 mM DTT, and 2 M urea. Cleaved protein samples were further purified by cation exchange chromatography using a buffer containing 50 mM Tris pH 7, 4 M Urea and a gradient of NaCl with concentrations spanning from 20 to 1000 mM and size exclusion chromatography where the samples were also exchanged into a final storage buffer containing 50 mM MES pH 6 and 4 M Gnd-HCl. The final samples were stored at 4 °C.

Prior to FCS and AUC experiments, A1-LCD was exchanged into a buffer containing 50 mM HEPES pH 7 with no denaturant. Care was taken to avoid introducing excess NaCl by using volatile acids and bases to adjust the pH. Buffer exchange was performed using 0.5 mL 7 kDa MWCO Zeba spin desalting columns (ThermoFisher). After desalting, samples were diluted 5-fold and concentrated twice to ensure complete buffer exchange. For SAXS experiments, A1-LCD samples were first exchanged into 1 M MES pH 6 buffer via 20× dilution and subsequent concentration; the procedure was repeated 3 times. The samples were then transferred into 50 mM HEPES pH 7 buffer by overnight dialysis.

**Small-angle X-ray scattering.** SAXS experiments were carried out at beamline 18ID-D (BioCAT) at the Advanced Photon Source at Argonne National Laboratory.

**Time-resolved SAXS.** In time-resolved SAXS experiments, the NaCl concentration in the A1-LCD sample was rapidly raised from zero (or rather no excess) NaCl to the desired concentration in the two-phase regime. Time-resolved experiments were performed using two separate microfluidic mixers to obtain information over a range of timescales. A chaotic-flow mixer was used to observe times from tens of microseconds to ~20 ms[69,70]. A laminar-flow mixer was used to observe times from ~1 to 80 ms[71]. For both chaotic and laminar-flow experiments, A1-LCD in 50 mM HEPES buffer was concentrated in Amicon centrifugal concentrators to ~20 mg/mL (~1.6 mM). The samples were diluted for laminar-flow experiments in 50 mM HEPES pH 7 buffer to 0.598 mM.

The laminar-flow microfluidic chip had 5 inlets. The central channel contained the protein sample. The diagonal channels contained a matched buffer taken from the dialysis reservoir. The vertical channels contained the matched buffer with the desired concentration of NaCl. NaCl-containing buffers were prepared such that they would not exceed the desired concentration by the end of the channel. Thus buffers were prepared at a concentration 9.8% greater than the maximum desired concentration. The maximum value is not achieved in the observed time window (Supplementary Fig. 13). The concentration increases steeply in the first several milliseconds, followed by a gradual increase approaching the maximum. During the measured window, the concentration varies between 88 and 92% of the maximum (Supplementary Fig. 13). The time dependence of the fractional increase is identical for all NaCl concentrations, and the NaCl ratio between different samples is constant across all time points. Thus, for simplicity, laminar-flow samples will be referred to by their final concentration at equilibrium.

The chaotic-flow microfluidic chip contained 3 inlets. The protein sample without excess salt was injected into the center channel. The two diagonal channels contained a matched buffer with the desired concentration of NaCl. The flow rate was set such that the protein sample would be diluted 5× into the NaCl buffer. A 20 mg/mL protein sample was used for a final concentration of 4 mg/mL in the measurement channel. The NaCl solution was prepared at 375 mM for a final concentration of 300 mM. Measurements were recorded by scanning across the channel at different distances from the point at which complete mixing is achieved. Buffer subtraction was done by injecting matched buffer into the center channel before and after the sample. The observable time points of chaotic-flow experiments ranged from 0.069 to 17 ms.

**Fluorescence correlation spectroscopy.** Fluctuations of the fluorescence intensity, as probed by fluorescence correlation spectroscopy (FCS), provide access to the diffusion time of fluorescent species. A1-LCD samples were labeled with CF660R dye (Biotium, California, USA) as previously reported[31]. Confocal fluorescence measurements were performed on a Picoquant MT200 instrument (Picoquant, Germany). Fluorophores were excited using a 640 nm pulsed laser (LDH-D-C-640, Picoquant, Germany) with a repetition rate of 20 MHz. Excitation power was measured before the objective with a laser photodiode and set to 8.0 μW for measurements in the dilute phase and 0.85 μW for measurements inside the dense phase. Emitted photons were collected with a ×60 1.2 UPlanSApo Superapochromat water immersion objective (Olympus, Japan), passed through a dichroic mirror (ZT470-488/640rpc, Chroma, USA), and filtered by a 100 μm pinhole (Thorlabs, USA). Photons were separated according to polarization using a polarizer beam splitter cube (Ealing, California, USA) and further refined by a filter H690/70 (Chroma, USA) in front of the SPAD detectors (Excelitas, USA). Photons are counted and accumulated by a HydraHarp 400 TCSPC module (Picoquant, Germany) with 1 ps resolution.

All measurements were performed in uncoated polymer coverslip cuvettes (Ibidi, Wisconsin, USA), significantly decreasing the fraction of protein adhering to the surface compared to normal glass cuvettes. For phase-separation measurements, 0.8 nM labeled A1-LCD was premixed with 400 μM unlabeled A1-LCD before the addition of NaCl.0. Reaction media was 50 mM Hepes, pH 7.0–7.1 (NH₄Cl), 0.002% v/v Tween20 and 200 mM β-mercaptoethanol. We verified that pH was not significantly affected when total NaCl was varied between 0 and 1 M. Mixing and measurements were performed at 22.5–23.5 °C.

To provide a precise calibration of the diffusion coefficient, we further perform df-FCS measurements of 1 nM Alexa 488-labeled A1-LCD. All measurements were performed at 22 °C. The dual focus configuration was realized on the same Microtime 200 confocal microscope using a differential interference contrast (DIC) prism before the objective and by alternating excitation of two orthogonally polarized diode lasers at 483 nm (LDH-D-C-485, PicoQuant) with a repetition rate of 40 MHz and a laser power of 30 μW each. The distance between the two foci was calibrated using the Alexa 488 (maleimide) dye diffusion coefficient as a reference standard. Experimental conditions were the same as stated above but in presence of 1 mM β-mercaptoethanol. Analysis of FCS traces is described in the Supporting Information.

All measurements were performed in uncoated polymer coverslip cuvettes (Ibidi, Wisconsin, USA), significantly decreasing the fraction of protein adhering to the surface compared to normal glass cuvettes. Labeled protein was added to a ratio of 1:1 × 10⁶ labeled to unlabeled protein.

**AUC.** Sedimentation velocity analytical ultracentrifugation (SV-AUC) experiments were carried out in a ProteomeLab instrument (Beckman Coulter, Indianapolis) equipped with Rayleigh interference optics, following standard protocols[72]. Samples were inserted in centerpieces with 3 or 12 mm optical pathlength, respectively, with H₂O as an optical reference, and temperature equilibrated to 20 °C in the rotor at rest, prior to acceleration to 50,000 rpm or 55,000 rpm. Interference data were acquired in 2 min intervals for 18 h and analyzed with a model of a sedimentation coefficient distribution c(s) with maximum entropy regularization at P = 0.99, and discrete cosolute signals[73,74] using the software SEDFIT. Nonideality corrections were applied to the sedimentation analysis at high concentrations in the absence of NaCl[75]. No nonideality corrections were applied to samples containing 50 mM NaCl.

The isotherm of integrated weight-average sedimentation coefficients was fit to different self-association models using the software SEDPHAT[76,77]. Monomer and dimer s-values were adjusted in nonlinear regression, and for isodesmic and two-step isodesmic models, sedimentation coefficients of higher oligomers were assumed to follow a 2/3-power hydrodynamic scaling law. Based on the best-fit equilibrium constants, the asymptotic sedimentation coefficient at the infinite time was calculated according to Gilbert's theory[41,77].

**Reporting summary.** Further information on research design is available in the Nature Research Reporting Summary linked to this article.

## Data availability
All data analyzed and data analysis scripts are available at https://github.com/erik-martin/LCD_nucleation. Source data are provided with this paper.

## Code availability
All custom code used in simulations is available at https://github.com/erik-martin/LCD_nucleation.

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

## Acknowledgements

T.M. acknowledges funding by NIH grant R01NS121114, the St. Jude Children's Research Hospital Research Collaborative on Membrane-less Organelles in Health and Disease, and by the American Lebanese Syrian Associated Charities. This work was supported by the Intramural Research Programs of the National Institute of Biomedical Imaging and Bioengineering, NIH. This research used resources of the Advanced Photon Source, a U.S. Department of Energy (DOE) Office of Science User Facility operated for the DOE Office of Science by Argonne National Laboratory under Contract No. DE-AC02-06CH11357 and resources supported by grant 9 P41 GM103622 from the National Institute of General Medical Sciences of the National Institutes of Health. Use of the

Pilatus 3 1M detector was provided by grant 1S10OD018090-01 from NIGMS. The content is solely the responsibility of the authors and does not necessarily reflect the official views of the National Institute of General Medical Sciences or the National Institutes of Health.

## Author contributions

E.W.M. and T.M. conceived of the project and designed the research. E.W.M prepared protein samples, performed and analyzed SAXS experiments, binodal measurements, and assisted with FCS measurements. T.S.H. designed, implemented and analyzed simulations of analytic nucleation. J.B.H. and S.C. provided technical and practical support for SAXS data collection and hardware design. J.B.H. performed the initial SAXS data reduction and averaging. J.J.I. and A.S. performed and analyzed FCS experiments. P.S. performed and analyzed AUC experiments. E.W.M. and T.M. wrote the manuscript. P.S., A.S., and T.M. acquired funding. All authors edited the manuscript.

## Competing interests

T.M. is a consultant for Faze Medicines, Inc. The affiliation has not influenced this work. The remaining authors declare no competing interests.
