## [Peer Review File · Nature Communications]

REVIEWER COMMENTS

Reviewer #1 (Remarks to the Author):

The manuscript by Martin et al. entitled "A multi-step nucleation process determines the kinetics of prion-like domain phase separation" addresses a significantly important kinetic issue using the low-complexity region of hnRNPA1 (A1-LCD) as an archetypal phase-separating protein. The authors utilize a unique amalgamation of methodologies involving time-resolved small-angle X-ray scattering (TR-SAXS), in conjunction with fluorescence correlation spectroscopy (FCS), sedimentation velocity analytical ultracentrifugation (SV-AUC), modeling and simulations to obtain novel molecular insights into the non-equilibrium process that results in the phase transition via nucleation from a mixed homogenous phase. A unique combination of turbulent-flow and laminar-flow microfluidic TR-SAXS measurements coupled with modeling using the Weibull probability distribution function allow them to determine how the quench depth governs the rate of nucleation in nonequilibrium measurements. The authors were able to discern the two distinct kinetic regimes involving the early molecular/nanoscale phenomena and the slower homogeneous nucleation mechanism at the mesoscale during the phase transition. This brilliant piece of work represents a technological tour de force in the field of liquid-liquid phase separation and will be of immense interest to the broad contemporary readership of Nature Communications. I have a few questions, comments, and suggestions for the authors to consider before the final publication.

1. Page 7 & Figure 2 C,D - FCS: A careful inspection of the FCS autocorrelation curves reveals something intriguing; the curves for the dilute phase appear to become faster as a function of increasing salt concentrations from 60 mM to 1 M NaCl (a smaller symbol size might be better). Does this indicate the salt-induced chain collapse resulting in the decrease in the average size? Of course, the viscosity correction needs to be invoked. I was wondering if the authors would like to include a plot of τ_D vs salt for the dilute phase as well (Figure 2D). In contrast, the FCS curves for the dense phase exhibit a progressively slower diffusion as a function of increasing ionic strength which is of course expected. Can the authors comment on the internal viscosity change in droplets as a function of the ionic strength using the Stokes-Einstein relationship (the sizes, RGs, are known under these conditions from their subsequent SAXS measurements)?

2. Page 7 & Figure 2 C,D: Also, it'd be interesting to know whether the translational diffusion coefficients estimated from the FCS in the dense phase correspond to a single A1-LCD diffusion or a diffusion of a cluster within the droplets. Also, my suggestion is to include texts "Dilute phase" and "Dense phase" next to the correlation curves in the figure to make it easier for the readers.

3. Page 8 & Figure 3B - Turbulent-flow TR-SAXS: The authors observe a decay (~ 470 microsec) and a growth (~ 36 ms) profile in their kinetic traces that are interpreted as salt-induced (barrier-limited) collapse followed by the assembly. They analyze the TR-SAXS curve by independently fitting exponential decay and growth curves (equations 1 & 2 in SI). They should indicate the SI equation numbers in Figure 3B legend. I was wondering if they would consider fitting using a single composite decay and growth function (as opposed to independently fitting using two equations) to analyze these data. My concern is that the fitting using two separate equations becomes a bit ambiguous in the intermediate time regime (~ 2-6 ms) where both decay and growth kinetics are operative. Additionally, the authors can state in the legend what the error bars mean. Are these from three independent measurements? Also, it might be better to change the time unit (abscissa) from sec to ms.

4. Page 8/9 & Figure 3: What's the molecular origin of the RG increase on a (slow) millisecond timescale? There are two possibilities. This component could arise either due to the changes in the average RG upon cluster formation or due to the single-chain expansions upon phase separation? Their previous work on coil-to-globule transitions (Ref 19; Biophys. J. 2020) showed that swapping of intrachain interactions for interchain interactions upon phase separation can potentially lead to chain expansion.

5. The form factor: It would be useful to state how the authors determined the form factor. They cite a

paper (Ref. 5 in SI, Riback et al. Science 2017) but it might be useful to discuss this in SI. The boxed region of the 18-ms plot in Figure 3C should also be explained in the manuscript. A few introductory statements on the form factor will also be useful for the later part on the time evolution of the form factor in laminar-flow TR-SAXS measurements (Figure 6).

6. Figure S4 (SI): What's the interpretation for the increase and decrease in the I0 as a function time? The authors state on page 8 of the manuscript "An increase in I0 is often associated with an increase in mass..." but it's not clear what's the reason for the observed change in their case. It's also interesting that the I0 is strikingly different (~ 1.4) for the aromatic-depleted LCD compared to A1-LCD (0.2-0.4). Any plausible explanation?

Also, notice in the legend of Figure S4 "Figure 4B" should be changed to "Figure 3B" in two places.

7. A few minor points:

(i) In the abstract, "protein architecture" can be changed to "sequence architecture" (or something similar), since for a broad audience, the "protein architecture" could mean the protein 3D structure.

(ii) The authors can consider including the amino acid sequence of A1-LCD in Figure 2 (Figure 2A?).

(iii) Page 6 (Results): The subheading "NaCl concentration controls the driving force..." can be changed to "Ionic strength controls the driving force...".

(iv) On page 7 the subheading "Millisecond timescale collapse of A1-LCD" can be changed to "Sub-millisecond timescale collapse of A1-LCD" since the authors are able to access a much faster sub-ms timescale with a deadtime of ~ 69 μ s.

(v) On page 10, the last line of the first paragraph - Figure 4C doesn't exist in the manuscript; did they mean Figure 4B?

(vi) On page 18, the last line in Discussion: The term "secondary nucleation" appears without any prior mention. What could be the secondary nucleation in this context?

(vii) I was wondering if the authors would like to also include the cumulative probability distribution expression (equation 9 on page 10 in SI) in the main paper on page 15 where they discuss the stretching exponent.

(viii) The authors can consider citing a relevant and comprehensive review article on nonequilibrium aspects of phase transitions by Berry, Brangwynne, and Haataja in Reports on Progress in Physics (2018) (<https://doi.org/10.1088/1361-6633/aaa61e>).

Reviewed by Samrat Mukhopadhyay

Reviewer #2 (Remarks to the Author):

The paper presents a detailed time-resolved SAXS study to reveal the kinetics and mechanism underlying of liquid-liquid phase separation (LLPS) of a prototypical prion-like domain induced by salt. A particular focus was to characterize the size distribution of clusters prior to phase separation. The results obtained from the rapid-mixing time-resolved SAXS approach are novel and interesting to the community. Some conclusions drawn need to be further substantiated however. Though SAXS is sensitive to structures on the nano- and mesoscale, an R_g value for such highly polydisperse system developing with time has a very limited meaning (is there a Guinier-region visible at all?). An interesting observation is that small complexes (oligomers) seem to form prior to the actual nucleation and growth process, which cannot be followed to completion by this technique, however. The kinetics of homogeneous nucleation and growth beyond the nano- and mesoscale could have been followed by time-resolved light scattering, which would allow to come up with mechanistic insights into the overall kinetics of the LLPS. Hence, in my view the statement that the insights presented here hold the key to understanding the vastly different timescales of phase separation of different biomolecules is not correct.

Further points: The mechanism of the salt-induced LLPS formation should be discussed (other than electrostatic screening?). The question regarding partial folding in the droplet phase could be answered by CD spectroscopy, I guess. The change in hydration density as reason for the change of the form factor could be checked as only one hydration sphere around the peptide has a slightly

higher scattering length density (typically about 3%). What is the meaning of an increase in stability of the hydration layer? As an interparticle correlation structure factor peak generally shows up as a broad maximum, the increase of R_g (which actually cannot be determined anymore beyond a few ms) is rather due to a rapid increase of larger assemblies. Finally, several other papers dealing with the kinetics of LLPS are neither mentioned nor discussed within the context of this work.

Reviewer #3 (Remarks to the Author):

Liquid-liquid phase separation is a hot topics, however little is known for its kinetic aspect. This manuscript address the time and space of how phase separation proceeds by the state-of-art rapid-mixing time-resolved small-angle scattering technique. Experimental data is well designed and solid, and its interpretation is sound.

I was deeply impressed by this work and the manuscript is wroth for publication in this journal.

Below is my minor concerns and suggestions which the author might consider.

Matching buffer subtraction:

In order to clarify mu understanding, in equilibrium inside the two phase regime droplets are formed in which assembly of biomolecule. Inside the droplet, the solvent condition is dense phase condition different from the dilute phase condition. Since the absorption of NaCl is substantial amount, the density profile appears to be like attached image. In chaotic-flow mixing, early events may comprise the development of clusters as well as the change in solvent density level. So, it would be helpful for understanding to show a schematics of density profile, which the author considers. The interpretation of the increase in I_0 may be too simplified.

Indication of size:

Since the manuscript deals with size, it is recommended to show the dimension more clearly. For example, the words, "nanoscale" and "mesoscale" may be ambiguous for readers of outside field. In present case the former corresponds to " a few nm" and the latter "< 100nm" (limited by angle region). In Figure 2.A the order of size of droplets is missing. (ca. 50 micro meter?) In Figure 9.c schematics each diagrams lack rough dimension.

Supplementary information:

What is the path length of the mixer ?

In Eqs (1) and (2), the square of R_g (R_g^2) is more appropriate than just R_g .

In Eq (3), $I(q)$ is divided by solute mass concentration?

The assembly metric appears to follow a Weibull probability distribution, regardless of q coverage range (<100 nm). Does it mean that the cluster associations larger than 100nm is different ?

regards,

Tetsuro Fujisawa

Reviewer #3 Attachment File:

Reviewer #4 (Remarks to the Author):

This is one of a series of recent papers from the Martin, Mittag lab. It is a follow up of a recent Science paper that reflects a collaboration with the Holehouse, Pappu labs. This submission represents a rather extensive effort to look at nucleation clusters after a jump to higher salt concentrations. In general, the presentation is interesting and complicated, requiring the reader to adjust to complex concepts. It would be useful for the general reader to define homogeneous (and heterogeneous) nucleation and its putative role in phase changes. Why is ice freezing similar to LLPS? How is it (as represented in Figure 9C) necessarily different?

On P 6 the authors write: "Our insights may explain why phase separation of different biomolecules happen over vastly different scales." To that end: Many LLPS systems have been shown to have stable oligomers. The intro refers to pathological aggregation as proceeding through distinct oligomers. An early example of the role of oligomers in LLPS is a paper by Li et al (45) with reversible oligomers of SH3 and polyproline sequences. AUC has been used by the Mittag lab, Correia lab and Casteneda lab to characterize the presence of stable, often isodesmic, oligomers. Micelles have also been observed in the ELP field (Chilkoti) that act as nuclei. None of this work seems to be referenced? Does the rapid mixing or quench approach, and the rapid formation of small oligomers and nuclei, only apply to systems that have subtle or low concentration nucleation events. Does this limit the insights to certain systems? Cells do not use salt quench (stress granules may) any more than they use temperature jumps.

There are general conclusions promoted through the paper: At the nanoscale small complexes form with low affinity? No numbers or fractions are presented. Raising the question how does AUC see transient complexes? At the mesoscale, monomers add with high affinity. But, how does AUC see mesoscale complexes that must pellet? Again no estimates of affinities are presented?

Specific comments about AUC: In general, the AUC data is used to complement the more extensive scattering, mixing data. But the AUC data is presented in a cavalier manner. The AUC presentation needs numbers. The data are pre-adjusted in the cNI function to remove the effect of k_s or thermodynamic nonideality. What is the value of k_s that is not being reported – I'm guessing at least ~20-30 ml/g? The results come from four data points with the highest concentration being bimodal.

Thus, the slope of the $1/s$ vs c plot not shown comes from 3 points? Nonideality usually dominates the high concentration data, so it appears odd that oligomers only grow out of the highest concentration data. What is K_2 for dimerization? Is this the low affinity step not reported? What is K_{iso} ? Is this the high affinity step not reported? Monomer addition is proposed; what role might ripening play in this process.

Nonideality does not broaden boundaries, it sharpens them. So the no salt explanation appears to be incorrect and represents oligomer formation. The bimodal distribution could also be and probably is kinetic. The explanation of a dimer step followed by an isodesmic step may reflect kinetics. Do those values also need slow kinetics to generate the distribution in 4A. The data points in the inserts in 4B are data from the upper panel of 4A; an equilibrium fit of 4 points determines the best fit, but I doubt this generates the bimodal shape. Direct boundary fitting with residuals would improve the story but at least report the best-fitted values. The text refers to Figures 4B,C but there is no 4C?

I take issue with the use of isodesmic but appreciate the tendency and need to simplify. Isodesmic refers to constant free energy of addition, which as represented in Figure 9 must actually be a distribution of contacts with different energies. Even isoenthalpic is probably too simple here. Thus, the authors state the low evolution of the structure factor, is unlikely to be diffusion limited, but why not dynamic conformational or contact changes that lead to assembly/nucleation? Is this not an origin of slow kinetics? The three steps in Figure 9C report k_1 , k_2 , k_3 but the forward arrows are large? Should k_1 have a larger back-step? The authors would benefit from looking at relaxation kinetics where off rates are essential to proper interpretation.

Other comments:

P 7 $csat$ is anticorrelated with NaCl lacks mechanism. You mean $csat$ decreases as NaCl concentration increases and suggests electrostatic repulsion as a simple mechanism. Next sentence says decorrelation time – should this be correlation time as the axis label in Fig 2C? Monomer R_g decreases with increasing NaCl concentration and “correlates” with $csat$. This usage seems nonmechanistic.

P 8 Rather than “results were in agreement with published results”, say what you mean, that the time constants are similar for cytochrome C dilution from denaturant and A1-LCD mixing with salt.

The symbols in Fig 3B are black and blue but not easily seen – open symbols, closed symbols.

P 9 – less solvent accessible surface means more stable hydration layer? Why not water release? How does that change contrast?

P 11 – “one order of magnitude greater than the uncertainty (Figure 5C not 5B?).”

Fig S9 – It is assumed that D is measured under dilute conditions (FCS) and not extrapolated to zero concentration? It would be useful to state conditions.

Figure S11 is Figure 8A – why is it reproduced here?

P 37 Figure 7C (and S12A) – no error bars on the average values. What are the dots in Figure S12B?

P 16 The size distribution of sub-critical clusters is determined by the nucleation barrier (Figure 1) and (insert is) therefore easiest to characterize near the binodal boundary.

P 17 - Alternately, biomolecules could be maintained in a supersaturated state if the nucleation barrier was high enough to prevent phase separation on relevant time scales, resulting (insert in) the system (delete to be) (insert being) kinetically trapped in the one-phase regime.

We thank the reviewers for their constructive criticism and insightful questions. We have made changes to the manuscript as laid out in the point-by-point response to specific reviewer comments below.

Reviewer #1

The manuscript by Martin et al. entitled "A multi-step nucleation process determines the kinetics of prion-like domain phase separation" addresses a significantly important kinetic issue using the low-complexity region of hnRNPA1 (A1-LCD) as an archetypal phase-separating protein. The authors utilize a unique amalgamation of methodologies involving time-resolved small-angle X-ray scattering (TR-SAXS), in conjunction with fluorescence correlation spectroscopy (FCS), sedimentation velocity analytical ultracentrifugation (SV-AUC), modeling and simulations to obtain novel molecular insights into the non-equilibrium process that results in the phase transition via nucleation from a mixed homogenous phase. A unique combination of turbulent-flow and laminar-flow microfluidic TR-SAXS measurements coupled with modeling using the Weibull probability distribution function allow them to determine how the quench depth governs the rate of nucleation in nonequilibrium measurements. The authors were able to discern the two distinct kinetic regimes involving the early molecular/nanoscale phenomena and the slower homogeneous nucleation mechanism at the mesoscale during the phase transition. This brilliant piece of work represents a technological tour de force in the field of liquid-liquid phase separation and will be of immense interest to the broad contemporary readership of Nature Communications.

We thank the Reviewer for their generous appreciation of our work.

I have a few questions, comments, and suggestions for the authors to consider before the final publication.

1.a Page 7 & Figure 2 C,D - FCS: A careful inspection of the FCS autocorrelation curves reveals something intriguing; the curves for the dilute phase appear to become faster as a function of increasing salt concentrations from 60 mM to 1 M NaCl (a smaller symbol size might be better). Does this indicate the salt-induced chain collapse resulting in the decrease in the average size? Of course, the viscosity correction needs to be invoked. I was wondering if the authors would like to include a plot of τ_D vs salt for the dilute phase as well (Figure 2D).

This is a good suggestion but the dependence of τ_D on salt concentration in the dilute phase is small; basically, only the data point at 60 mM NaCl differs from the others. The FCS experiments were designed to measure concentrations in the dilute and dense phase, which means that the measurements in the dilute phase are at the sensitivity limit of the method. The labeled protein is least dilute at 60 mM NaCl (because the saturation concentration is highest), and we find this data point most reliable. We thus prefer not to interpret the extent of τ_D differences between samples, while the 2f-FCS experiments shown confirm the low dependence of τ_D on the salt concentration.

1.b In contrast, the FCS curves for the dense phase exhibit a progressively slower diffusion as a function of increasing ionic strength which is of course expected. Can the authors comment on the internal viscosity change in droplets as a function of the ionic strength using the Stokes-Einstein relationship (the sizes, R_G s, are known under these conditions from their subsequent SAXS measurements)?

We do not know the ionic strength in the dilute and dense phases at a given total salt concentration, and thus cannot properly adjust for their effects on viscosity. To address the reviewer's suggestion, we have included an apparent viscosity, i.e., the τ_D normalized by τ_D at the lowest salt concentration, in Figure 2D. We observe an increase in the apparent viscosity by

4-fold in the dense phase. However, the dense phase concentration does not change, pointing to stronger intermolecular interactions. We have clarified this on pg. 7:

“The apparent viscosity increases but the protein concentration in the dense phase is largely constant, pointing to stronger intermolecular interactions in the resulting dense phase, i.e., stronger networking.”

2. Page 7 & Figure 2 C,D: Also, it'd be interesting to know whether the translational diffusion coefficients estimated from the FCS in the dense phase correspond to a single A1-LCD diffusion or a diffusion of a cluster within the droplets.

The brightness per diffusing molecule is constant across the two phases and the salt concentrations, i.e., our measurements are consistent with a single labeled molecule in the measurement volume. We cannot tell how many unlabeled protein molecules are associated with the labeled molecule and we are thus hesitant to overinterpret our data through an empirical model. In fact, this limitation motivated the use of the trSAXS experiments in which we were able to observe clustering and assembly.

Also, my suggestion is to include texts "Dilute phase" and "Dense phase" next to the correlation curves in the figure to make it easier for the readers.

Thank you for the useful suggestion. We have included labels for the dilute and dense phase to the figure.

3. Page 8 & Figure 3B - Turbulent-flow TR-SAXS: The authors observe a decay (~ 470 microsec) and a growth (~ 36 ms) profile in their kinetic traces that are interpreted as salt-induced (barrier-limited) collapse followed by the assembly. They analyze the TR-SAXS curve by independently fitting exponential decay and growth curves (equations 1 & 2 in SI). They should indicate the SI equation numbers in Figure 3B legend. I was wondering if they would consider fitting using a single composite decay and growth function (as opposed to independently fitting using two equations) to analyze these data. My concern is that the fitting using two separate equations becomes a bit ambiguous in the intermediate time regime (~ 2-6 ms) where both decay and growth kinetics are operative.

We thank the reviewer for this useful suggestion. We have updated the fitting equation to be the sum of the exponential decay and growth. The equation has been changed in the SI and the description updated to:

“The evolution of R_G^2 with time was modeled using an equation with the sum of single exponential decay and growth regimes:

$$R_G^2 = A_1 e^{-\frac{t}{\tau_1}} + A_2 \left(1 - e^{-\frac{t}{\tau_2}}\right) + C_1 \quad (1)$$

The growth regime was modeled as $1 - e^{-\frac{t}{\tau_2}}$ because the data appeared to converge toward a metastable cluster size distribution prior to nucleation.”

The figure caption now says

“The solid line is a fit to the sum of exponential collapse ($\tau \sim 470 \mu\text{s}$) and growth ($\tau \sim 36 \text{ ms}$) [Eq. 1 in SI].”

Additionally, the authors can state in the legend what the error bars mean. Are these from three independent measurements? Also, it might be better to change the time unit (abscissa) from sec to ms.

We have changed the time unit on the abscissa to ms. With respect to the error values, multiple measurements were all averaged into one SAXS curve per data point. The error from each individual measurement was propagated to the final curve. Error bars displayed in the figure

represent the uncertainty in determining the radius based on the form factor fit, and the experimental error is considered here. To clarify this, we have added:

SI Methods:

“Uncertainty was then propagated through radial averaging and all subsequent operations on the 1D profiles including averaging, subtraction and binning in q space, using standard addition in quadrature. All methods used were implemented in the BioXTAS RAW software and represent the standard treatment of uncertainty for SAXS data.”

Figure Caption:

“Error bars represent the uncertainty in the fit to the IDR form factor.”

4. Page 8/9 & Figure 3: What's the molecular origin of the R_G increase on a (slow) millisecond timescale? There are two possibilities. This component could arise either due to the changes in the average R_G upon cluster formation or due to the single-chain expansions upon phase separation? Their previous work on coil-to-globule transitions (Ref 19; Biophys. J. 2020) showed that swapping of intrachain interactions for interchain interactions upon phase separation can potentially lead to chain expansion.

As the reviewer rightly points out, we would indeed expect chain expansion in the dense phase compared to the dilute phase. In the context of Figure 3, we are observing the formation of clusters. We can say this confidently due to the evolution in the shape of the SAXS form factor. Additionally, once clusters begin to form, our SAXS measurements do no longer report on single monomers but on the clusters. This point is clarified in the revised text (page 9): “The appearance of this upturn (Figure 3D, box) demonstrates that the apparent increase in R_G is due to assembly and not simply an expansion of a single chain.”

5. The form factor: It would be useful to state how the authors determined the form factor. They cite a paper (Ref. 5 in SI) but it might be useful to discuss this in SI. The boxed region of the 18-ms plot in Figure 3C should also be explained in the manuscript. A few introductory statements on the form factor will also be useful for the later part on the time evolution of the form factor in laminar-flow TR-SAXS measurements (Figure 6).

The form factor was empirically derived from poly-alanine simulations in the cited reference (Riback et al. 2017), and we have also used this approach in other papers (Martin et al. 2020; Martin, Hopkins, and Mittag 2021; Martin et al. 2021; Bremer et al. 2021). This method of fitting SAXS data was specifically parameterized to work well with unfolded proteins and is fairly robust to the presence of small amounts of inter-particle interference. We understand this method is not yet widely used so have added more detail to the SI to clarify.

“The form factor was empirically derived by Riback et al. by simulating poly-alanine peptides of varying length and $C\beta$ attractive potential. The resulting form factor is an interpolation between dimensionless calculated SAXS profiles where the only fitting parameters are R_G and the Flory scaling exponent.”

6. Figure S4 (SI): What's the interpretation for the increase and decrease in the I_0 as a function time? The authors state on page 8 of the manuscript "An increase in I_0 is often associated with an increase in mass..." but it's not clear what's the reason for the observed change in their case.

The I_0 is a function of the concentration and M_w of the molecules in solution. We found the increase in I_0 accompanying the decrease in R_g to be surprising as well. However, similar behavior was observed in several independent measurements of A1-LCD and did not appear in any of our controls. Therefore, we concluded that the inverse relationship between R_g and I_0 is a feature unique to this sequence. Unfortunately, our SAXS experiments were not well poised to

interrogate this behavior in detail, so at present we are left to speculate about its origins. After considering multiple potential sources, we hypothesized that differential hydration of extended disordered proteins versus compact disordered proteins could cause this behavior. We modeled how an increase in solvation density around compact conformations could result in both a decrease in R_g and an increase in I_0 and included this as a supplementary note. We stress that this is only a potential explanation and hope this hypothesis motivates future studies that use experimental methods better suited to studying hydration.

It's also interesting that the I_0 is strikingly different (~ 1.4) for the aromatic-depleted LCD compared to A1-LCD (0.2-0.4). Any plausible explanation?

Yes: the A1-LCD and aro-depleted LCD were measured on two different days. The intensity is not in absolute units and therefore I_0 is dependent on the normalization specific to measurements on a given day. In this particular case, there was a large difference in flux between the measurements for the two constructs. We have clarified this in the text in the figure caption: "Measurements of A1-LCD and the aromatic depleted LCD were performed on different days with different x-ray flux. I_0 is not given in absolute units, so the values cannot be directly compared between samples." Additionally, we have added the unit (AU) to the I_0 axis in Figure S4.

Also, notice in the legend of Figure S4 "Figure 4B" should be changed to "Figure 3B" in two places.

Thank you for taking note! We have corrected this error.

7. A few minor points:

(i) In the abstract, "protein architecture" can be changed to "sequence architecture" (or something similar), since for a broad audience, the "protein architecture" could mean the protein 3D structure.

Thank you for the suggestion. We have changed this to read "protein amino acid sequence."

(ii) The authors can consider including the amino acid sequence of A1-LCD in Figure 2 (Figure 2A?).

We have now included this sequence in Figure 3. We agree with the suggestion to include information about the exact sequence and thought that it might be even more helpful to show it where data from two different LCD sequences are being directly compared.

(iii) Page 6 (Results): The subheading "NaCl concentration controls the driving force..." can be changed to "Ionic strength controls the driving force..."

Thank you for this suggestion. We have made the change.

(iv) On page 7 the subheading "Millisecond timescale collapse of A1-LCD" can be changed to "Sub-millisecond timescale collapse of A1-LCD" since the authors are able to access a much faster sub-ms timescale with a deadtime of $\sim 69 \mu\text{s}$.

Thank you. We have made this change.

(v) On page 10, the last line of the first paragraph - Figure 4C doesn't exist in the manuscript; did they mean Figure 4B?

Thank you. This has been corrected.

(vi) On page 18, the last line in Discussion: The term "secondary nucleation" appears without any prior mention. What could be the secondary nucleation in this context?

Thank you for noting this confusing term. We had used this term to mean nucleation of phase separation after either formation of oligomers or a conformational change, but it is unneeded jargon and we have thus removed it:

“Our results highlight that cells may regulate phase separation not just by modulating the saturation concentration but also by modulating the formation of complexes on pathway to phase separation and hence modifying the rate of nucleation.”

(vii) I was wondering if the authors would like to also include the cumulative probability distribution expression (equation 9 on page 10 in SI) in the main paper on page 15 where they discuss the stretching exponent.

This is a valid suggestion, but we chose not to include the equations in the main text because we did not want to draw attention to the specific form of the probability distribution. In fact, multiple distributions would likely fit the data and provide near identical results. We also wanted to avoid confusion with the well-known Avrami (JMAK) equation which has a near identical form to the Weibull CDF but describes a fundamentally different kinetic process – both the nucleation and growth of crystals. We think it is important not to overinterpret the specific form of the distribution. The critical point is simply that the data are best fit to a probability distribution where the probability of nucleation increases with time denoting a second underlying process.

(viii) The authors can consider citing a relevant and comprehensive review article on nonequilibrium aspects of phase transitions by Berry, Brangwynne, and Haataja in Reports on Progress in Physics (2018) (<https://doi.org/10.1088/1361-6633/aaa61e>).

Thank you for this good suggestion. We have included this reference.

Reviewed by Samrat Mukhopadhyay

Reviewer #2

The paper presents a detailed time-resolved SAXS study to reveal the kinetics and mechanism underlying of liquid-liquid phase separation (LLPS) of a prototypical prion-like domain induced by salt. A particular focus was to characterize the size distribution of clusters prior to phase separation. The results obtained from the rapid-mixing time-resolved SAXS approach are novel and interesting to the community. Some conclusions drawn need to be further substantiated however. Though SAXS is sensitive to structures on the nano- and mesoscale, an R_g value for such highly polydisperse system developing with time has a very limited meaning (is there a Guinier-region visible at all?). An interesting observation is that small complexes (oligomers) seem to form prior to the actual nucleation and growth process, which cannot be followed to completion by this technique, however. The kinetics of homogeneous nucleation and growth beyond the nano- and mesoscale could have been followed by time-resolved light scattering, which would allow to come up with mechanistic insights into the overall kinetics of the LLPS. Hence, in my view the statement that the insights presented here hold the key to understanding the vastly different timescales of phase separation of different biomolecules is not correct.

We thank the reviewer for their insight. Philosophically, we agree with their point. Our work focuses on two specific time scales that both fall in the micro- to millisecond range. We observe an initial collapse followed by clustering and nucleation of phase separation. Our work was intended to extend to shorter time scales versus what has been reported for the time evolution of phase separated systems (i.e., droplet growth by Ostwald ripening or coalescence). We propose that our work synergizes with such studies to provide insight into vastly different timescales.

To clarify our intent, we have modified the abstract as follows:

“We find two kinetic regimes **on the micro- to millisecond timescale** that are distinguished by the size distribution of clusters.”

“The work we present highlights that careful multi-pronged characterization is required for the understanding of mechanisms of condensate assembly and lays out a clear path towards understanding how the kinetics of biological phase separation is encoded in biomolecules.”

With respect to following the SAXS measured R_g as a function of time, we took care to only report radii when they are meaningful. As the reviewer suggests, this is only true when the radii are within the measurable length scales and there is a reasonable Guinier region. We note in Figure 3D that reporting an R_g becomes complex due to deviations from the monomer form factor at small angles at 18 ms. We have now explicitly stated this in Figure caption 3D: “The boxed region at 18 ms highlights the departure of the form factor at small angles from a monomer and thus indicates beginning assembly, **beyond which point we stop reporting R_g values.**” Further, the use of an assembly metric to monitor mesoscopic assembly was implemented specifically because R_g was no longer meaningful in these solutions.

Further points: The mechanism of the salt-induced LLPS formation should be discussed (other than electrostatic screening?).

The precise mechanism of NaCl-induced phase separation of prion-like LCDs is still a subject of active investigation (as we discussed recently in (Martin et al. 2021)). Here, we exploited this salt-dependence to induce phase separation and manipulate quench depth in a manner that is experimentally practical, but the mechanism was not the target of our investigation.

If we were to speculate based on the current state of research on the topic, the salt dependence originates from two contributions: screening of electrostatic repulsion and the hydrophobic effect; we expect them to contribute differently to the change in c_{sat} at different salt concentrations. The bulk of the change in c_{sat} occurs between 0-400 mM NaCl. At 400 mM NaCl, the Debye length is ~ 0.5 nm and on a similar order to the size of a single amino acid. The role of electrostatic screening is further supported by the significant repulsive interactions observed in AUC experiments in the absence of salt. In the DDX4 system, in which NaCl has the inverse effect on phase separation, the majority of the salt effect could be modeled using a screened Coulombic potential (Nott et al. 2015). In addition, increasing salt will increase the solvation energy of hydrophobic residues favoring their interaction. This effect is expected to contribute little at low salt concentrations and be the main effect at high salt concentrations (Krainer et al. 2021).

Given that the discussion above is somewhat speculative, we have expanded our discussion and pointed to relevant literature on the topic:

“The precise mechanism of how salt impacts phase separation is a subject of active study and is likely protein dependent. In the case of A1-LCD, we suspect NaCl screens repulsive interactions originating from the net positive charge and promotes distributive interactions between aromatic residues.”

The question regarding partial folding in the droplet phase could be answered by CD spectroscopy, I guess.

The SAXS form factor can already exclude partial folding. We see no changes in the shape at higher angles indicating that the LCD is still behaving as an unfolded polymer.

The change in hydration density as reason for the change of the form factor could be checked as only one hydration sphere around the peptide has a slightly higher scattering length density (typically about 3%). What is the meaning of an increase in stability of the hydration layer?

This is a great question, which we have explored in the supplementary note. By "increased stability", we mean that in a compact state a disordered protein could trap more water molecules in a semi-stable hydration shell and thus the density would increase. This is built on the premise that disordered proteins, particularly extended disordered proteins, appear to have a hydration layer that is significantly less dense than folded proteins (Henriques et al. 2018). As also mentioned in response to Reviewer 1, this hypothesis remains to be rigorously tested and is, by no means, the only possible explanation. We have clarified this explanation:

"We instead suggest that compact conformations of disordered proteins may have more stable hydration layers. While a collapsed IDR has less solvent accessible area, we posit an increase in stability of the hydration layer which would, in turn, increase contrast and I_0 while decreasing R_G (Figure S5, Supplementary Note)."

As an interparticle correlation structure factor peak generally shows up as a broad maximum, the increase of R_G (which actually cannot be determined anymore beyond a few ms) is rather due to a rapid increase of larger assemblies.

We agree with the Reviewer and have changed the wording to clarify this:

"Given the slow evolution of the form factor and the fact that large assemblies do not appear until ~18 ms after mixing..."

Finally, several other papers dealing with the kinetics of LLPS are neither mentioned nor discussed within the context of this work.

We have now cited:

In reference to general theory:

(Berry, Brangwynne, and Haataja 2018) (as also suggested by Reviewer 1)

In reference to cluster formation and phase separation in synthetic ELP systems:

(Lyons et al. 2014), (Zai-Rose et al. 2018) and (Hassouneh et al. 2015).

In reference to cluster formation and phase separation from high concentration globular protein solutions:

(Zhang et al. 2012), (Pan, Vekilov, and Lubchenko 2010) and (Pan et al. 2007).

In reference to observing nucleation and growth after quenching into the two-phase regime:

(Cinar and Winter 2020).

Reviewer #3:

Liquid-liquid phase separation is a hot topics, however little is known for its kinetic aspect. This manuscript address the time and space of how phase separation proceeds by the state-of-art rapid-mixing time-resolved small-angle scattering technique. Experimental data is well designed and solid, and its interpretation is sound.

I was deeply impressed by this work and the manuscript is wroth for publication in this journal.

Below is my minor concerns and suggestions which the author might consider.

Matching buffer subtraction:

In order to clarify mu understanding, in equilibrium inside the two phase regime droplets are

formed in which assembly of biomolecule. Inside the droplet, the solvent condition is dense phase condition different from the dilute phase condition. Since the absorption of NaCl is substantial amount, the density profile appears to be like attached image.

In chaotic-flow mixing, early events may comprise the development of clusters as well as the change in solvent density level. So, it would be helpful for understanding to show a schematics of density profile, which the author considers. The interpretation of the increase in I_0 may be too simplified.

We thank the reviewer for their comments. Indeed, we also considered ion adsorption as a potential source for the changes in I_0 and it could be a contributing factor. We chose to focus on the hydration shell because we do not know how NaCl partitions in the dilute and dense phases. In reality, it is difficult to deconvolute impacts from ion adsorption and hydration shell density in our data but changes in contrast could stem from both. To clarify this, we have included text in the Supplemental Note to indicate the possible effects from ions.

“It is important to note that this effect is due to an increase in excess contrast from the solvation shell. If an additional increase in contrast arises from ion adsorption or an enhanced ionic double layer, this would also be consistent with the models.”

The density of the solvent versus the density of mesoscopic assemblies (dense phase droplets) is not relevant for the chaotic-flow measurements because at nearly all time points there is a well-defined Guinier region and thus only small clusters of molecules exist in solution. It is possible that this could have an effect toward the end of the chaotic flow experiments, but the clusters are still small, and the volume fraction of protein is only ~ 0.001 which would limit the impact of edge effects on the bulk solution.

This issue becomes more intriguing in the laminar flow mixing experiments. We do not believe there is an actual difference in buffer density in these experiments. Na and Cl ions are free to flow into the sample at all time points and it is reasonable to assume that small changes in ion density would rapidly be compensated for by diffusion. Additionally, it is far from certain that ions will partition into the dense phase as predicted by Voorn-Overbeek theory. Indeed, as noted by the work of Charles Sing and Sarah Perry (among others), the opposite case, where the excluded volume of the salt and polymers cause salt to partition into the dilute phase, is often true (Radhakrishna et al. 2017). In absence of precise measurements of salt partitioning (which have not yet been reported for phase separation of LCDs), we have to rely on the naïve simplification that Na and Cl concentrations are effectively equal, and the solvent density is unaffected.

That said, Dr. Fujisawa's diagram could be interpreted as surface effects where there is a locally low concentration of ions at the interface between the dilute and dense phases. If this were the case, it would not significantly alter the analysis of nucleation kinetics but could be a reason that the boundary between dilute and dense phases appears very well defined (as indicated by the power law decay of $d = 3.8$ which is more characteristic of a smooth versus a fuzzy surface). In light of the uncertainty in ion densities inside and outside of the protein dense phase and the minimal impact on the extracted kinetic parameters (where I_0 is normalized) we feel that adding this information would distract from the data. However, the points are well-taken and specific effects of ion and ion partitioning are the subject of active projects.

Indication of size:

Since the manuscript deals with size, it is recommended to show the dimension more clearly. For example, the words, "nanoscale" and "mesoscale" may be ambiguous for readers of outside field. In present case the former corresponds to "a few nm" and the latter "< 100nm" (limited by angle region).

Thank you for the helpful suggestion. We have clarified these scales by defining nanoscale in the abstract and including the following text in the introduction:

“How nucleation of protein phase separation occurs on molecular length scales from single molecules to small oligomers, i.e., the nanoscale, and on the order of clusters of many molecules with radii over 100 nm, i.e., the mesoscale, is thus an interesting question.”

In Figure 2.A the order of size of droplets is missing. (ca. 50 micro meter?)

We have added a scale bar to Figure 2A.

In Figure 9.c schematics each diagrams lack rough dimension.

Thank you for pointing out the lack of scale. A rough dimension has been added at each step in Figure 9C.

Supplementary information:

What is the path length of the mixer ?

Thank you for pointing out this omission. The path length is 1.025 mm for the laminar flow mixer and 0.25 mm for the chaotic flow mixer. We have added this information to the Supplemental Methods.

In Eqs (1) and (2), the square of R_g (R_g^2) is more appropriate than just R_g .

The reviewer is correct, thank you for pointing out this error. Equations 1 and 2 in the SI have been combined in response to Reviewer 1, the change to the square of R_g has been implemented and the fit recalculated. Figures 3 and S4 have been updated accordingly.

In Eq (3), $I(q)$ is divided by solute mass concentration?

The concentration is effectively treated as the volume fraction divided by the volume of an individual molecule. In the model, the volume fraction is neglected as all conditions have the same volume and volume fraction. The mass difference between the conditions is accounted for in the difference in densities (ρ). This has now been clarified in the text:

“The total volume of the core-shell sphere is held constant and thus the simulated data neglects ϕ . Differences in mass are accounted for in the contrast term, ρ .”

The assembly metric appears to follow a Weibull probability distribution, regardless of q coverage range (<100 nm). Does it mean that the cluster associations larger than 100nm is

different ?

This is a very interesting point. Would we expect different kinetics for associations that occur between clusters >100 nm? Our expectation is that there is a second correlation length that is outside of our accessible range. Thus far attempts to measure these systems using USAXS have not yielded reasonable signal, but this effort is ongoing. However, we would speculate that this correlation length may probe the kinetic regime of droplet coarsening and not nucleation. The kinetics of coarsening would be on the time scale of minutes to hours and follow a power law of $t^{1/3}$. Similar studies have been done on polymer systems and on globular proteins using USAXS/USANS in the minutes to hours range and have seen the expected coarsening behavior. We appreciate this comment and do believe that monitoring this evolution using USAXS/USANS coupled to optical microscopy would be a very interesting follow-up story compared to the much earlier regime we are probing in the current manuscript.

*regards,
Tetsuro Fujisawa*

Reviewer #4

This is one of a series of recent papers from the Martin, Mittag lab. It is a follow up of a recent Science paper that reflects a collaboration with the Holehouse, Pappu labs. This submission represents a rather extensive effort to look at nucleation clusters after a jump to higher salt concentrations. In general, the presentation is interesting and complicated, requiring the reader to adjust to complex concepts. It would be useful for the general reader to define homogeneous (and heterogeneous) nucleation and it's putative role in phase changes. Why is ice freezing similar to LLPS? How is it (as represented in Figure 9C) necessarily different?

We thank the reviewer for their thoughtful comment. We have included text in the Introduction to make exactly the suggested comparison: "Phase separation in the metastable region requires nucleation (Figure 1) **akin to supercooled water that does not freeze until a nucleated.**" In the Discussion, we now say: "Solutions near the binodal may remain a single phase indefinitely in the absence of perturbations that aid nucleation, **similar to supercooled water. The unfavorable initial association into small clusters differs from the analogy to freezing water.**"

On P 6 the authors write: "Our insights may explain why phase separation of different biomolecules happen over vastly different scales." To that end: Many LLPS systems have been shown to have stable oligomers. The intro refers to pathological aggregation as proceeding through distinct oligomers. An early example of the role of oligomers in LLPS is a paper by Li et al (45) with reversible oligomers of SH3 and polyproline sequences. AUC has been used by the Mittag lab, Correia lab and Casteneda lab to characterize the presence of stable, often isodesmic, oligomers. Micelles have also been observed in the ELP field (Chilkoti) that act as nuclei. None of this work seems to be referenced?

We erroneously omitted referencing work on ELPs, which was foundational in the field. We have added appropriate references to work from the Correia and Chilkoti groups.

The reviewer mentions that oligomers have been observed for domain/motif-type systems such as multivalent SH3/polyproline, in the seminal publication by Mike Rosen's lab (Li et al. 2012) or also our own work with SPOP. However, the current manuscript focuses on early steps in phase separation of intrinsically disordered LCDs, and we thus did not cite that work.

Does the rapid mixing or quench approach, and the rapid formation of small oligomers and nuclei, only apply to systems that have subtle or low concentration nucleation events. Does this

limit the insights to certain systems? Cells do not use salt quench (stress granules may) any more than they use temperature jumps.

This is an interesting point. We, of course, use a rapid quench approach to isolate the early steps in nucleation. Nucleation in cells will likely proceed via many different mechanisms. What our work indicates is that nucleation rates are the product of both the thermodynamics driving forces for phase separation AND early oligomerization steps that are distinct. We suggest that modulating these early steps is an efficient mechanism for cells to regulate phase separation when, as the reviewer correctly points out, there is actually no salt or temperature quench. We think the true power of these methods is that they are easily transferrable to many systems. Indeed, systems with slower kinetics, such as those that proceed through micellization, would be even more information-rich as the pre-nuclei species could be resolved well on SAXS length scales. Per the Reviewer's suggestion, we have further highlighted the transferability of these methods.

“Combining rapid mixing TR-SAXS measurements with careful equilibrium characterization of oligomeric species has the potential to uncover the on-pathway steps to LLPS nucleation in complex mixtures where greater deviations from homogeneous nucleation can be expected.”

There are general conclusions promoted through the paper: At the nanoscale small complexes form with low affinity? No numbers or fractions are presented. Raising the question how does AUC see transient complexes? At the mesoscale, monomers add with high affinity. But, how does AUC see mesoscale complexes that must pellet? Again no estimates of affinities are presented?

The intention of the manuscript was to highlight the mechanism of association without invoking specific affinities. This was a conscious decision because, as the reviewer points out, our experimental approach using an ionic strength quench is expected to yield results that are mechanistically similar to biologically relevant processes if not identical in magnitude. To clear up this confusion, we have added the extracted K_d values to Figure 4.

Regarding the second half of the question: The reviewer is of course correct in pointing to the fact that large assemblies rapidly pellet. We note this in the text: “Any dense phase droplets that formed sedimented rapidly under these conditions.” The AUC data focused on small oligomers. We cannot observe these distributions by AUC at concentrations above the saturation concentration, so statements about the addition of monomers at the mesoscale are inferred from the mechanism of small oligomer formation. The conclusions we draw about the rate of mesoscale assembly formation arise from TR-SAXS experiments, and we simply state that when observed at the mesoscale, nucleation appears to follow a classical nucleation model. This is noted on page 15: “Taken together, we conclude that A1-LCD phase separation seems to follow a homogeneous nucleation pathway when observed on the mesoscale.” In the abstract, the process of early oligomer formation and nucleation might be confusing, so we have rewritten the sentence: **“After the initial unfavorable complex assembly, additional monomers are added with higher affinity. At the mesoscale, assembly resembles classical homogeneous nucleation.”**

Specific comments about AUC: In general, the AUC data is used to complement the more extensive scattering, mixing data. But the AUC data is presented in a cavalier manner. The AUC presentation needs numbers. The data are pre-adjusted in the cNI function to remove the effect of k_s or thermodynamic nonideality. What is the value of k_s that is not being reported – I'm guessing at least ~20-30 ml/g?

The reviewer's guess is very close, and we have included this information in the revised text.

“In the absence of excess NaCl, A1-LCD exhibited strongly nonideal sedimentation (best estimate $k_s = 28$ mL/g) with little indication of oligomerization (Figure 4A).”

The results come from four data points with the highest concentration being bimodal. Thus, the slope of the $1/s$ vs c plot not shown comes from 3 points? Nonideality usually dominates the high concentration data, so it appears odd that oligomers only grow out of the highest concentration data.

In the presence of 50 mM salt, the data show significantly reduced nonideality due to the screening of charge repulsion. Therefore, its effect was not separately accounted for and standard $c(s)$ analysis was used in this case. To highlight this, we have extended the sentence: “In contrast, sedimentation of samples containing 50 mM NaCl showed **dramatically decreased nonideality and a clear** concentration-dependent shift in the sedimentation velocity distribution peaks, characteristic of self-association (Figure 4A).”

We have also fixed a typo in the axis label of figure 4A which now reads “ $c(s)$ ” instead of $c_{NI}(s)$. Finally, we have added the sentence to the Methods:

“No nonideality corrections were applied to samples containing 50 mM NaCl.”

What is K_2 for dimerization? Is this the low affinity step not reported? What is K_{iso} ? Is this the high affinity step not reported? Monomer addition is proposed; what role might ripening play in this process.

We have added the K_d values to Figure 4. Ripening will play a role but will arise in a different kinetic regime which will last from minutes to hours. Our TR-SAXS data effectively “count” the number of nucleated droplets. This is evident in the stochasticity that is shown in Figure 7D,E,F. These results are effectively blind to ripening. Ripening would be evident in the slow evolution of correlation lengths longer than our measurable length scale (>100 nm). These effects typically have a characteristic power law behavior of $t^{1/3}$. It is clear that our data reports on nucleation and not ripening.

Nonideality does not broaden boundaries, it sharpens them. So the no salt explanation appears to be incorrect and represents oligomer formation.

The reviewer is correct, and we have deleted the incorrect sentence.

The bimodal distribution could also be and probably is kinetic. The explanation of a dimer step followed by an isodesmic step may reflect kinetics. Do those values also need slow kinetics to generate the distribution in 4A. The data points in the inserts in 4B are data from the upper panel of 4A; an equilibrium fit of 4 points determines the best fit, but I doubt this generates the bimodal shape. Direct boundary fitting with residuals would improve the story but at least report the best-fitted values.

We believe that kinetics are unlikely to play a role in the observed distributions. It has been well documented that SV is sensitive to kinetic effects only when complex lifetimes are on the timescale of sedimentation, i.e., on the scale of hours. In a back-of-the-envelope calculation, with K_d values in the mM range, such lifetimes would require on-rate constants on the order of $k_{on} = k_{off}/K_d = (1e-3/sec)/1e-3M = 1/Msec$. This is many orders of magnitude lower than even extremely slow protein-protein interactions. Of note, TR-SAXS measurements monitor processes on the millisecond timescale.

Direct boundary fitting would, in our opinion, distract from the salient physical phenomena we are aiming to unravel qualitatively. It would not add more quantitation than the sw isotherm analysis already shown, for which we have now provided best-fit K_d values, considering also that the binding models we chose were to distinguish qualitative behavior of self-association

schemes, and are by no means certain. The bimodal nature of the sedimentation coefficient distributions are well in line with previous results from Gilbert's theory (famously concluding the existence of bimodal boundaries in (Gilbert 1959; Schuck and Zhao 2017) and their mathematical calculation leaves no room for judgement.

The text refers to Figures 4B,C but there is no 4C?

Thank you. We have corrected this typo.

I take issue with the use of isodesmic but appreciate the tendency and need to simplify. Isodesmic refers to constant free energy of addition, which as represented in Figure 9 must actually be a distribution of contacts with different energies. Even isoenthalpic is probably too simple here. Thus, the authors state the low evolution of the structure factor, is unlikely to be diffusion limited, but why not dynamic conformational or contact changes that lead to assembly/nucleation? Is this not an origin of slow kinetics?

We agree completely with the Reviewer's assessment of an isodesmic association model describing the formation of critical nuclei. Our intent in using this model was to describe the initial formation of multimers containing only a small number of monomers. If our experiments were sensitive to the formation of larger assemblies approaching the critical nuclei size, the isodesmic model would almost certainly be invalid. However, we found it to be quite valuable in describing the early stages of cluster formation. In this sense we were able to clearly observe the formation of a low affinity dimer before higher affinity addition of monomers. We don't want to give the impression that we are overinterpreting the isodesmic model, so we have included the following sentence in the main text:

"To characterize the formation of small assemblies that form on-pathway to critical nuclei formation, we assumed that simple association models would be valid in the limit of clusters containing only a few molecules."

The three steps in Figure 9C report k_1 , k_2 , k_3 but the forward arrows are large? Should k_1 have a larger back-step? The authors would benefit from looking at relaxation kinetics where off rates are essential to proper interpretation.

We thank the reviewer for the suggestion. A larger k_{-1} arrow would indeed better represent our proposed model. This has been modified in the figure. We have commented above regarding the lack of needing to take into account kinetics in the interpretation of our AUC data.

Other comments:

P 7 c_{sat} is anticorrelated with NaCl lacks mechanism. You mean c_{sat} decreases as NaCl concentration increases and suggests electrostatic repulsion as a simple mechanism. Monomer R_g decreases with increasing NaCl concentration and "correlates" with c_{sat} . This usage seems nonmechanistic.

The precise mechanism of NaCl induced phase separation of prion-like LCDs is still a subject of active investigation as we discussed recently (Martin et al. 2021). In the current work, we exploited this salt dependence to induce phase separation and manipulate quench depth in a manner that is experimentally practical, but the mechanism was not a target of our investigation.

If we were to speculate based on the current state of research on the topic, the salt dependence originates from two contributions: screening of electrostatic repulsion and the hydrophobic effect; we expect them to contribute differently to the change in c_{sat} at different salt concentrations. The bulk of the change in c_{sat} occurs between 0-400 mM NaCl. At 400 mM NaCl, the Debye length is ~ 0.5 nm and on a similar order to the size of a single amino acid. The role of electrostatic screening is further supported by the significant repulsive interactions

observed in AUC experiments in the absence of salt. In the DDX4 system, in which NaCl has the inverse effect on phase separation, the majority of the salt effect could be modeled using a screened Coulombic potential (Nott et al. 2015). In addition, increasing salt will increase the solvation energy of hydrophobic residues favoring their interaction. This effect is expected to contribute little at low salt concentrations and be the main effect at high salt concentrations (Krainer et al. 2021).

Given that the discussion above is somewhat speculative, we have expanded our discussion and pointed to relevant literature on the topic:

“The precise mechanism of how salt impacts phase separation is a subject of active study and is likely protein dependent. In the case of A1-LCD, we suspect NaCl screens repulsive interactions originating from the net positive charge and promotes distributive interactions between aromatic residues.”

Next sentence says decorrelation time – should this be correlation time as the axis label in Fig 2C?

The use of “correlation time” in the x-axis of the FCS curves is a somewhat confusing convention of FCS. To avoid this confusion, we have simplified the axis title to “lag time”.

P 8 Rather than "results were in agreement with published results", say what you mean, that the time constants are similar for cytochrome C dilution from denaturant and A1-LCD mixing with salt.

To ensure that the mixer is producing reproducible, reliable results, all measurements were preceded by measurements of refolding of cytochrome c from denaturant – a process with well-established kinetics. The fact that the kinetics of cytochrome c refolding and A1-LCD collapse have similar time constants is simply a combination of coincidence and some similarity in the two processes. We have clarified this in the text to avoid confusion. It now reads:

“We used the same chaotic-flow mixer to record the response of cytochrome c to dilution of denaturant, a well characterized refolding process, to ensure the mixer was properly functioning. The measured time constants were consistent with the established kinetics of refolding (Figure S2) (Kathuria et al. 2014). Interestingly, the time constant of A1-LCD collapse is of the same order as the initial step in cytochrome c which has been attributed to a barrier limited collapse, suggesting that the reorganization of A1-LCD to more compact conformations might also be barrier limited.”

The symbols in Fig 3B are black and blue but not easily seen – open symbols, closed symbols. We have changed the blue to open symbols to increase the color difference.

P 9 – less solvent accessible surface means more stable hydration layer? Why not water release? How does that change contrast?

A folded protein would have less solvent accessible area than an unfolded protein. However, the important parameter in SAXS is the density of the solvation shell. Disordered proteins have a dynamic solvation layer. We are suggesting that the hydration shell is more stable in collapsed conformations which could result in it effectively becoming thicker or denser (as referenced above). Importantly, our modeling shows that very small changes in density can have a large impact on R_g / l_0 . In response to this suggestion and that of other reviewers, we have clarified our meaning:

“We instead suggest that compact conformations of disordered proteins may have more stable hydration layers. While a collapsed IDR has less solvent accessible area, we posit an increase in stability of the hydration layer which would, in turn, increase contrast and l_0 while decreasing

R_c (Figure S5, Supplementary Note).”

P 11 – “one order of magnitude greater than the uncertainty (Figure 5C not 5B?).”

Thank you for catching this, it has been corrected.

Fig S9 – *It is assumed that D is measured under dilute conditions (FCS) and not extrapolated to zero concentration? It would be useful to state conditions.*

These measurements were performed under dilute conditions. We have added the specific concentration to the Methods and Figure S9.

“Experiments were carried out on samples of 1 nM Alexa488-labeled A1-LCD.”

Figure S11 is Figure 8A – why is it reproduced here?

There is a minor difference between Figures S11 and 8A. For clarity, Figure 8A shows the average from multiple simulations at the same ionic strength. Figure S11 shows the independent traces for each simulation. Figure S11 was included to show the unaveraged data.

P 37 Figure 7C (and S12A) – no error bars on the average values. What are the dots in Figure S12B?

The figures do not include error bars because they are all below the size of the marker. Since the assembly metric relies only on the highest intensity data, it can be calculated with great precision. We chose not to include error estimates because it is our opinion that these remarkably small values do not represent the true uncertainty in the values which will certainly contain contributions from things like the quality of the normalization that is exceptionally difficult to quantify and propagate to individual data points. We have now clarified this in the methods.

“Given the high scattering intensity at the smallest angles and that the assembly metric is the average of 39 data points, **the values can be calculated with extreme precision and the uncertainty is below the size of the markers in all datasets. There are other potential sources of uncertainty that are difficult to quantify, such as the quality of the normalization, so we chose to exclude mention of uncertainty in the figures. Any additional sources of uncertainty are likely small as well due to the high signal to noise in the region of the curves that is analyzed. Therefore, we conclude that** the variation in the assembly metric between measurements at a given time point is resultant from real variations in the volume fraction of assembled protein and not simply a lack in precision in the data.”

The dots in figure S12B are the experimental data points. We have corrected this omission from the figure legend: “**The blue circles represent the experimentally determined heterogeneity.**”

P 16 The size distribution of sub-critical clusters is determined by the nucleation barrier (Figure 1) and (insert is) therefore easiest to characterize near the binodal boundary.

Thank you, this was revised.

P 17 - Alternately, biomolecules could be maintained in a supersaturated state if the nucleation barrier was high enough to prevent phase separation on relevant time scales, resulting (insert in) the system (delete to be) (insert being) kinetically trapped in the one-phase regime.

Thank you for the suggestion. This has been changed.

Point-by-point References

- Berry, J., C. P. Brangwynne, and M. Haataja. 2018. 'Physical principles of intracellular organization via active and passive phase transitions', *Rep Prog Phys*, 81: 046601.
- Bremer, Anne, Mina Farag, Wade M. Borchers, Ivan Peran, Erik W. Martin, Rohit V. Pappu, and Tanja Mittag. 2021. 'Deciphering how naturally occurring sequence features impact the phase behaviors of disordered prion-like domains', *bioRxiv*: 2021.01.01.425046.
- Cinar, H., and R. Winter. 2020. 'The effects of cosolutes and crowding on the kinetics of protein condensate formation based on liquid-liquid phase separation: a pressure-jump relaxation study', *Sci Rep*, 10: 17245.
- Gilbert, Geoffrey Alan. 1959. 'Sedimentation and electrophoresis of interacting substances. I. Idealized boundary shape for a single substance aggregating reversibly', *Proc. R. Soc. Lond. A*, 250: 377-88.
- Hassouneh, W., E. B. Zhulina, A. Chilkoti, and M. Rubinstein. 2015. 'Elastin-like Polypeptide Diblock Copolymers Self-Assemble into Weak Micelles', *Macromolecules*, 48: 4183-95.
- Henriques, J., L. Arleth, K. Lindorff-Larsen, and M. Skepo. 2018. 'On the Calculation of SAXS Profiles of Folded and Intrinsically Disordered Proteins from Computer Simulations', *J Mol Biol*, 430: 2521-39.
- Kathuria, S. V., C. Kayatekin, R. Barrea, E. Kondrashkina, R. Graceffa, L. Guo, R. P. Nobrega, S. Chakravarthy, C. R. Matthews, T. C. Irving, and O. Bilsel. 2014. 'Microsecond barrier-limited chain collapse observed by time-resolved FRET and SAXS', *J Mol Biol*, 426: 1980-94.
- Krainer, G., T. J. Welsh, J. A. Joseph, J. R. Espinosa, S. Wittmann, E. de Csillery, A. Sridhar, Z. Toprakcioglu, G. Gudiskyte, M. A. Czekalska, W. E. Arter, J. Guillen-Boixet, T. M. Franzmann, S. Qamar, P. S. George-Hyslop, A. A. Hyman, R. Collepardo-Guevara, S. Alberti, and T. P. J. Knowles. 2021. 'Reentrant liquid condensate phase of proteins is stabilized by hydrophobic and non-ionic interactions', *Nat Commun*, 12: 1085.
- Li, P., S. Banjade, H. C. Cheng, S. Kim, B. Chen, L. Guo, M. Llaguno, J. V. Hollingsworth, D. S. King, S. F. Banani, P. S. Russo, Q. X. Jiang, B. T. Nixon, and M. K. Rosen. 2012. 'Phase transitions in the assembly of multivalent signalling proteins', *Nature*, 483: 336-40.
- Lyons, D. F., V. Le, W. H. Kramer, G. L. Bidwell, 3rd, E. A. Lewis, D. Raucher, and J. J. Correia. 2014. 'Effect of basic cell-penetrating peptides on the structural, thermodynamic, and hydrodynamic properties of a novel drug delivery vector, ELP[V5G3A2-150]', *Biochemistry*, 53: 1081-91.
- Martin, E. W., A. S. Holehouse, I. Peran, M. Farag, J. J. Incicco, A. Bremer, C. R. Grace, A. Soranno, R. V. Pappu, and T. Mittag. 2020. 'Valence and patterning of aromatic residues determine the phase behavior of prion-like domains', *Science*, 367: 694-99.
- Martin, E. W., J. B. Hopkins, and T. Mittag. 2021. 'Small-angle X-ray scattering experiments of monodisperse intrinsically disordered protein samples close to the solubility limit', *Methods Enzymol*, 646: 185-222.
- Martin, Erik W., F. Emil Thomasen, Nicole M. Milkovic, Matthew J. Cuneo, Christy R Grace, Amanda Nourse, Kresten Lindorff-Larsen, and Tanja Mittag. 2021. 'Interplay of folded domains and the disordered low-complexity domain in mediating hnRNPA1 phase separation', *Nucleic Acids Research*, 49: 2931-45.
- Nott, T. J., E. Petsalaki, P. Farber, D. Jervis, E. Fussner, A. Plochowietz, T. D. Craggs, D. P. Bazett-Jones, T. Pawson, J. D. Forman-Kay, and A. J. Baldwin. 2015. 'Phase transition of a disordered nuage protein generates environmentally responsive membraneless organelles', *Mol Cell*, 57: 936-47.
- Pan, W., O. Galkin, L. Filobelo, R. L. Nagel, and P. G. Vekilov. 2007. 'Metastable mesoscopic clusters in solutions of sickle-cell hemoglobin', *Biophys J*, 92: 267-77.
- Pan, W., P. G. Vekilov, and V. Lubchenko. 2010. 'Origin of anomalous mesoscopic phases in protein solutions', *J Phys Chem B*, 114: 7620-30.

- Radhakrishna, Mithun, Kush Basu, Yalin Liu, Rasmia Shamsi, Sarah L. Perry, and Charles E. Sing. 2017. 'Molecular Connectivity and Correlation Effects on Polymer Coacervation', *Macromolecules*, 50: 3030-37.
- Riback, J. A., M. A. Bowman, A. M. Zmyslowski, C. R. Knoverek, J. M. Jumper, J. R. Hinshaw, E. B. Kaye, K. F. Freed, P. L. Clark, and T. R. Sosnick. 2017. 'Innovative scattering analysis shows that hydrophobic disordered proteins are expanded in water', *Science*, 358: 238-41.
- Schuck, Peter, and H. Zhao. 2017. *Sedimentation Velocity Analytical Ultracentrifugation: Interacting Systems* (CRC Press).
- Zai-Rose, V., S. J. West, W. H. Kramer, G. R. Bishop, E. A. Lewis, and J. J. Correia. 2018. 'Effects of Doxorubicin on the Liquid-Liquid Phase Change Properties of Elastin-Like Polypeptides', *Biophys J*, 115: 1431-44.
- Zhang, Fajun, Felix Roosen-Runge, Andrea Sauter, Roland Roth, Maximilian W. A. Skoda, Robert M. J. Jacobs, Michael Sztucki, and Frank Schreiber. 2012. 'The role of cluster formation and metastable liquid—liquid phase separation in protein crystallization', *Faraday Discussions*, 159: 313-25.

REVIEWER COMMENTS

Reviewer #1 (Remarks to the Author):

The authors have responded to all my questions and comments. The revised manuscript can now be accepted for publication in Nature Communications.

Regards,
Samrat Mukhopadhyay

Reviewer #3 (Remarks to the Author):

The authors properly addressed my questions. The article would greatly affect future SAXS experiments.

Reviewer #4 (Remarks to the Author):

The response and revisions answer most of my questions.

Reviewer 1 asked for a change from salt to ionic strength. LLPS often depends upon the surface tension of cluster formation; surface tension also strongly depends upon salt concentration and closely follows the Hofmeister series. Thus, the term salt dependent should not be changed to ionic strength, since ionic strength does not convey the correct message. This aspect of salt dependence is independent of protein charge but does not exclude compaction due to charge screening. The revision on page 7 should be modified accordingly.

P 16 “a mosaic of semi-stable contacts” – I think mosaic is the wrong term – “ensemble of interconverting conformations”. Mosaics do not interconvert.

Reviewer #1 (Remarks to the Author):

The authors have responded to all my questions and comments. The revised manuscript can now be accepted for publication in Nature Communications.

Regards,
Samrat Mukhopadhyay

Thank you for the helpful, constructive review.

Reviewer #3 (Remarks to the Author):

The authors properly addressed my questions. The article would greatly affect future SAXS experiments.

Thank you for the comments and review.

Reviewer #4 (Remarks to the Author):

The response and revisions answer most of my questions.

Thank you for the suggestions. They improved the final manuscript.

Reviewer 1 asked for a change from salt to ionic strength. LLPS often depends upon the surface tension of cluster formation; surface tension also strongly depends upon salt concentration and closely follows the Hofmeister series. Thus, the term salt dependent should not be changed to ionic strength, since ionic strength does not convey the correct message. This aspect of salt dependence is independent of protein charge but does not exclude compaction due to charge screening. The revision on page 7 should be modified accordingly.

We agree. This is an important point to note and we appreciate the reviewer bringing it to our attention. We believe that, at the salt concentrations used, the primary effect stems from the increase of the ionic strength. However, additional effects that are sensitive to the nature of the salt used become dominant as the concentration is increased. Indeed, Murthy et al¹ have observed this in the LC domain of FUS. This particular protein is less charged than the hnRNPA1 LCD, but it highlights the importance of the point. In order to give the reader a more nuanced view of salt interactions, we have specified “salt concentration” instead of “ionic strength”. We have additionally augmented the text to read:

“In the case of A1-LCD, we suspect that increasing ionic strength screens repulsive interactions originating from the net positive charge. As the salt concentration increases further, the enhancement of hydrophobic interactions will promote distributive interactions between aromatic residues. These effects will depend on the specific salt type.”

P 16 “a mosaic of semi-stable contacts” – I think mosaic is the wrong term – “ensemble of interconverting conformations”. Mosaics do not interconvert.

Thank you for the suggestion. We have made this change.

1 Murthy, A. C. *et al.* Molecular interactions underlying liquid-liquid phase separation of the FUS low-complexity domain. *Nat Struct Mol Biol* **26**, 637-648, doi:10.1038/s41594-019-0250-x (2019)